# Hierarchical Semi-Implicit Variational Inference with Application to Diffusion Model Acceleration

**Longlin Yu**[1,*]**, Tianyu Xie**[1,*]**, Yu Zhu**[3,4,*]**, Tong Yang**[5]**, Xiangyu Zhang**[5]**, Cheng Zhang**[1,2,†]

[1] School of Mathematical Sciences, Peking University
[2] Center for Statistical Science, Peking University
[3] Institute of Automation, Chinese Academy of Sciences
[4] Beijing Academy of Artificial Intelligence
[5] Megvii Technology Inc.
{llyu, tianyuxie}@pku.edu.cn, zhuyu2022@ia.ac.cn,
{yangtong, zhangxiangyu}@megvii.com, chengzhang@math.pku.edu.cn

## Abstract

Semi-implicit variational inference (SIVI) has been introduced to expand the analytical variational families by defining expressive semi-implicit distributions in a hierarchical manner. However, the single-layer architecture commonly used in current SIVI methods can be insufficient when the target posterior has complicated structures. In this paper, we propose hierarchical semi-implicit variational inference, called HSIVI, which generalizes SIVI to allow more expressive multi-layer construction of semi-implicit distributions. By introducing auxiliary distributions that interpolate between a simple base distribution and the target distribution, the conditional layers can be trained by progressively matching these auxiliary distributions one layer after another. Moreover, given pre-trained score networks, HSIVI can be used to accelerate the sampling process of diffusion models with the score matching objective. We show that HSIVI significantly enhances the expressiveness of SIVI on several Bayesian inference problems with complicated target distributions. When used for diffusion model acceleration, we show that HSIVI can produce high quality samples comparable to or better than the existing fast diffusion model based samplers with a small number of function evaluations on various datasets.

## 1 Introduction

Variational inference (VI) is an approximate Bayesian inference method that is gaining in popularity, where one tries to find an approximation to the target posterior distribution using an optimization approach (Jordan et al., 1999; Wainwright & Jordan, 2008; Blei et al., 2016). To do that, it first posits a family of variational distributions and then seeks the closest member from this family that minimizes some statistical distance to the target posterior, usually the Kullback-Leibler (KL) divergence. As the posterior is not analytically available, an equivalent formulation is often adopted in practice where one maximizes the evidence lower bound (ELBO) instead (Jordan et al., 1999).

One classical VI method is mean-field VI, which assumes a factorizable structure of the variational distributions over the parameters or latent variables (Bishop & Tipping, 2000). This often leads to closed-form coordinate-ascent update rules when certain conditional conjugacy conditions are satisfied. In practice, the conditional conjugacy may not hold and the true posterior could be much

---

[*]Equal contribution. This work was done during an internship at Megvii Technology Inc.
[†]Corresponding Author.

more complicated than what a factorized variational distribution can accurately approximate. In recent years, several attempts have been made in VI that alleviate these constraints by designing more flexible variational families (Jaakkola & Jordan, 1998; Saul & Jordan, 1996; Giordano et al., 2015; Tran et al., 2015; Rezende & Mohamed, 2015; Dinh et al., 2017; Kingma et al., 2016; Papamakarios et al., 2019), together with generic training algorithms via Monte Carlo gradient estimators (Nott et al., 2012; Paisley et al., 2012; Ranganath et al., 2014; Rezende et al., 2014; Kingma & Welling, 2014). While successful, these approaches all assume tractable densities of variational distributions. To further expand the capacity of variational families, one approach is to incorporate the implicit models that have intractable densities but are easy to sample from (Huszár, 2017; Tran et al., 2017; Mescheder et al., 2017; Shi et al., 2018a,b; Song et al., 2019). However, as the densities are intractable for implicit models, one often resorts to density ratio estimation for ELBO evaluation during training, which is known to be difficult in high dimensional settings (Sugiyama et al., 2012). To avoid density ratio estimation, semi-implicit variational inference (SIVI) has been proposed where the variational distributions are formed through a semi-implicit hierarchical construction, and various training criteria have been employed (Yin & Zhou, 2018; Moens et al., 2021; Titsias & Ruiz, 2019; Yu & Zhang, 2023).

While striking a good balance between approximation flexibility and training efficiency, current SIVI methods often use a single conditional layer which can be insufficient when the target posterior possesses complicated structures (e.g., multimodality, see an example in Section 5.1). To enhance the expressiveness of single-layer models, an intuitive but effective approach is to extend them to multi-layer hierarchical models (Vahdat & Kautz, 2020; Ranganath et al., 2016; Sobolev & Vetrov, 2019). In this paper, we propose hierarchical semi-implicit variational inference (HSIVI), which is a generalization of SIVI that allows multiple conditional layers. Instead of training the hierarchical semi-implicit model end to end, we introduce auxiliary distributions that interpolate between a simple base distribution and the target distribution to guide the intermediate semi-implicit distributions toward the target distribution. The conditional layers are then trained sequentially to match these auxiliary bridging distributions given the fitted semi-implicit distributions from the previous layers (Figure 1), using different criteria from before. This way, HSIVI allows progressive learning of the target distribution that significantly reduces the burden of each conditional layer. Moreover, HSIVI with the score matching objective can also be used to accelerate the sampling process of diffusion models where the pre-trained score networks corresponding to different noise levels provide a natural sequence of bridging distributions. In experiments, we demonstrate the effectiveness of HSIVI on both Bayesian inference tasks with complicated target distributions and diffusion model acceleration.

## 2 Background on semi-implicit variational inference

The semi-implicit variational family (Yin & Zhou, 2018; Titsias & Ruiz, 2019) is defined as

$$q_\phi(\boldsymbol{x}) = \int q_\phi(\boldsymbol{x}|\boldsymbol{z})q(\boldsymbol{z})\mathrm{d}\boldsymbol{z}, \qquad (1)$$

where $\phi$ are the variational parameters, $q_\phi(\boldsymbol{x}|\boldsymbol{z})$ is called the conditional layer, and $q(\boldsymbol{z})$ is called the mixing layer. This variational family is said to be semi-implicit as $q_\phi(\boldsymbol{x}|\boldsymbol{z})$ is required to be explicit and $q(\boldsymbol{z})$ is often implicit. The semi-implicit variational family is capable of capturing more complicated dependencies between variables (Yin & Zhou, 2018; Titsias & Ruiz, 2019; Yu & Zhang, 2023) than explicit variational families without the hierarchical structure. Given the observed data $D$, the classical VI methods often use the evidence lower bound (ELBO) for training, which is defined as $\mathrm{ELBO} := \mathbb{E}_{q_\phi(\boldsymbol{x})}\left[\log p(D, \boldsymbol{x}) - \log q_\phi(\boldsymbol{x})\right]$. However, as $q_\phi(\boldsymbol{x})$ is no longer tractable in SIVI, alternative training objectives have been introduced.

**ELBO related objectives**    Yin & Zhou (2018) considered a sequence of lower bounds of the ELBO

$$\mathcal{L}_{\text{SIVI-LB}}(p(\boldsymbol{x}|D)\|q_\phi(\boldsymbol{x})) := \mathbb{E}_{\boldsymbol{z}\sim q(\boldsymbol{z}), \boldsymbol{x}\sim q_\phi(\boldsymbol{x}, \boldsymbol{z})}\mathbb{E}_{\{\boldsymbol{z}^{(i)}\}_{i=1}^{K}\overset{\text{i.i.d.}}{\sim}q(\boldsymbol{z})}\log\frac{p(D, \boldsymbol{x})}{\frac{1}{K+1}\left(q_\phi(\boldsymbol{x}|\boldsymbol{z}) + \sum_{k=1}^{K}q_\phi(\boldsymbol{x}|\boldsymbol{z}^{(k)})\right)}.$$
$$(2)$$

It is an asymptotically exact surrogate in the sense that $\lim_{K\to\infty}\mathcal{L}_{\text{SIVI-LB}} = \text{ELBO}$. Titsias & Ruiz (2019) proposed unbiased implicit variational inference (UIVI) which uses samples from the inverse conditional distribution $q_\phi(\boldsymbol{z}|\boldsymbol{x})$ (from an MCMC run, e.g. Hamiltonian Monte Carlo (Neal, 2011)) to provide an unbiased gradient estimator of the exact ELBO. See more details of UIVI in Appendix B.

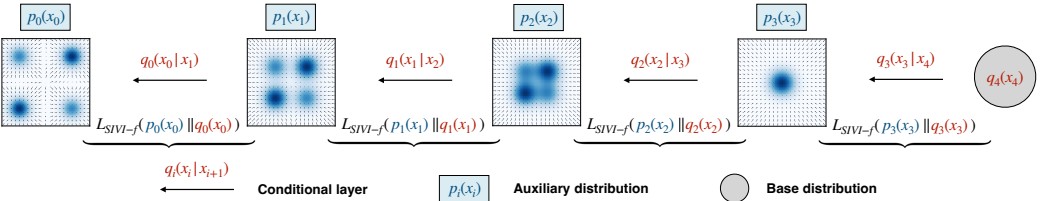

Figure 1: An example for 4-layer HSIVI. The target distribution $p_0(\boldsymbol{x})$ is a Gaussian mixture and the auxiliary distributions $\{p_i(\boldsymbol{x})\}_{i=0}^3$ are constructed using the diffusion bridge. The auxiliary distributions are plotted in the squares, where the blue heatmap describes the probability density and the arrows represent the score functions of the auxiliary distributions.

**Score matching objective** Besides the ELBO, score based distance measures have also been used for variational inference where the score function $\boldsymbol{S}(\boldsymbol{x}) := \nabla_{\boldsymbol{x}} \log p(\boldsymbol{x}|D) = \nabla_{\boldsymbol{x}} \log p(D, \boldsymbol{x})$ is assumed to be tractable (Liu et al., 2016; Zhang et al., 2018; Hu et al., 2018). Yu & Zhang (2023) considered the following Fisher divergence between the target distribution and the semi-implicit variational distribution

$$\mathcal{D}_{\text{Fisher}}(p(\boldsymbol{x}|D)\|q_\phi(\boldsymbol{x})) := \mathbb{E}_{\boldsymbol{x} \sim q_\phi(\boldsymbol{x})}\|\boldsymbol{S}(\boldsymbol{x}) - \nabla_{\boldsymbol{x}} \log q_\phi(\boldsymbol{x})\|_2^2. \tag{3}$$

By reformulating $\mathcal{D}_{\text{Fisher}}$ as the maximum of the following optimization problem

$$\mathcal{D}_{\text{Fisher}}(p(\boldsymbol{x}|D)\|q_\phi(\boldsymbol{x})) = \max_{\boldsymbol{f}(\boldsymbol{x})} \left[2\boldsymbol{f}(\boldsymbol{x})^T(\boldsymbol{S}(\boldsymbol{x}) - \nabla_{\boldsymbol{x}} \log q_\phi(\boldsymbol{x})) - \|\boldsymbol{f}(\boldsymbol{x})\|_2^2\right],$$

and using a similar trick as in denoising score matching (Vincent, 2011; Song & Ermon, 2019), one can transform the minimization of $\mathcal{D}_{\text{Fisher}}$ into the following minimax problem which is tractable

$$\min_\phi \max_\psi \mathcal{L}_{\text{SIVI-SM}}(p(\boldsymbol{x}|D)\|q_\phi(\boldsymbol{x})) := \mathbb{E}_{\boldsymbol{z} \sim q(\boldsymbol{z}), \boldsymbol{x} \sim q_\phi(\boldsymbol{x}|\boldsymbol{z})} \left[2\boldsymbol{f}_\psi(\boldsymbol{x})^T[\boldsymbol{S}(\boldsymbol{x}) - \nabla_{\boldsymbol{x}} \log q_\phi(\boldsymbol{x}|\boldsymbol{z})] - \|\boldsymbol{f}_\psi(\boldsymbol{x})\|_2^2\right]. \tag{4}$$

In practice, $\boldsymbol{f}_\psi(\boldsymbol{x})$ is parametrized using neural networks. The above minimax optimization problem can be efficiently solved by optimizing $\psi$ and $\phi$ alternately.

## 3 Hierarchical semi-implicit variational inference

The semi-implicit variational family $q_\phi(\boldsymbol{x})$ in equation (1) is indeed a single-layer model in the sense that it contains only one conditional layer. Our main idea is to expand this single-layer semi-implicit variational family into its multi-layer variants and introduce a sequence of auxiliary distributions to guide the semi-implicit distributions toward the target distribution. This leads to a new SIVI method which we call hierarchical semi-implicit variational inference (HSIVI). We start with the following definition which is motivated by equation (1).

**Definition 1** (Hierarchical Semi-Implicit Distribution). *Let $\boldsymbol{x}_T \sim q_T(\boldsymbol{x}_T)$ for some $T \in \mathbb{N}^*$, where $q_T(\boldsymbol{x}_T)$ is called the variational prior. Let $q_t(\boldsymbol{x}_t|\boldsymbol{x}_{t+1}; \phi_t)$ be the $t$-th conditional layer for $0 \le t \le T-1$. Denote $\{\phi_k\}_{k=t}^{T-1}$ by $\phi_{\ge t}$. The $t$-th layer hierarchical semi-implicit distribution $q_t(\boldsymbol{x}_t; \phi_{\ge t})$ is defined recursively from $T-1$ to $0$ by*

$$q_t(\boldsymbol{x}_t; \phi_{\ge t}) = \int q_t(\boldsymbol{x}_t|\boldsymbol{x}_{t+1}; \phi_t) q_{t+1}(\boldsymbol{x}_{t+1}; \phi_{\ge t+1}) \mathrm{d}\boldsymbol{x}_{t+1}, \quad 0 \le t \le T-1, \tag{5}$$

*where $q_T(\boldsymbol{x}_T; \phi_{\ge T}) := q_T(\boldsymbol{x}_T)$. Here, the $t$-th conditional layer $q_t(\boldsymbol{x}_t|\boldsymbol{x}_{t+1}; \phi_t)$ is required to be explicit and reparametrizable with a tractable score function $\nabla_{\boldsymbol{x}_t} \log q_t(\boldsymbol{x}_t|\boldsymbol{x}_{t+1}; \phi_t)$.*

Compared to the single-layer semi-implicit variational family (1), the family of hierarchical semi-implicit distributions provides a principled way to construct more expressive mixing layers using multi-layer architectures. Also, unlike the hierarchical variational models (Ranganath et al., 2016) which require an extra reverse model and explicit variational prior, hierarchical semi-implicit distributions inherit the advantage of SIVI that allows $q_t(\boldsymbol{x}_t; \phi_{\ge t})$ to be implicit, and as shown next, they do not require a reverse model and can be progressively trained using the simple algorithms of SIVI for each conditional layer, from $t = T-1$ to $t = 0$.

---

**Algorithm 1** Hierarchical semi-implicit variational inference (sequential training)

---

**Input:** Auxiliary bridge $\{p_t(\boldsymbol{x})\}_{t=0}^{T-1}$; initial value of parameters $\phi^{(0)} = \{\phi_i^{(0)}\}_{t=0}^{T-1}$.
**Output:** The optimal parameters $\phi^*$.
Initialization: $\phi \leftarrow \phi^{(0)}$.
**for** $t = T-1$ **to** $0$ **do**
    **while** not converge **do**
        Sample a minibatch $\{\boldsymbol{x}_T^{(k)}\}_{k=1}^K$ from the variational prior $q_T(\boldsymbol{x}_T)$.
        **if** $t < T-1$ **then**
            Sequentially sample $\{\boldsymbol{x}_{t+1}^{(k)}\}_{k=1}^K$ through $q(\boldsymbol{x}_i|\boldsymbol{x}_{i+1}; \phi_i)$ from $i = T-1$ to $i = t+1$.
            Detach the computation graphs from $\{\boldsymbol{x}_{t+1}^{(k)}\}_{k=1}^K$.
        **end if**
        Update $\phi_t$ by optimizing the $\mathcal{L}_{\text{SIVI-}f}\left(p_t(\boldsymbol{x}_t)\|q_t(\boldsymbol{x}_t; \phi_{\geq t})\right)$ based on the minibatch $\{\boldsymbol{x}_{t+1}^{(k)}\}_{k=1}^K$.
    **end while**
    $\phi_t^* \leftarrow \phi_t$.
**end for**
$\phi^* \leftarrow \{\phi_t^*\}_{t=0}^{T-1}$.

---

### 3.1 Progressive approximation with the auxiliary bridge

In this section, we introduce a bridging technique for progressively approximating the target distribution $p(\boldsymbol{x})$ using hierarchical semi-implicit distributions. Rather than approximating $p(\boldsymbol{x})$ with $q_0(\boldsymbol{x}; \phi_{\geq 0})$ directly, we construct a sequence of intermediate auxiliary distributions $\{p_t(\boldsymbol{x})\}_{t=0}^{T-1}$ as a bridge between the target distribution $p_0(\boldsymbol{x}) := p(\boldsymbol{x})$ and an easy-to-approximate distribution $p_{T-1}(\boldsymbol{x})$, to amortize the difficulty of one-pass fitting. A typical example of an auxiliary bridge is the geometric interpolation as described below.

**Example 1** (Geometric Interpolation). *Let $\boldsymbol{S}(\boldsymbol{x}) := \nabla \log p(\boldsymbol{x})$ be the score function of target distribution $p(\boldsymbol{x})$ and $\boldsymbol{S}_{base}(\boldsymbol{x}) := \nabla \log p_{base}(\boldsymbol{x})$ be the score function of a base distribution $p_{base}(\boldsymbol{x})$. In geometric interpolation (Neal, 2001; Bernton et al., 2019), each auxiliary distribution $p_t(\boldsymbol{x})$ for $0 \leq t \leq T-1$ has the following probability density function (pdf) and score function*

$$p_t(\boldsymbol{x}) \propto p_{base}(\boldsymbol{x})^{1-\lambda_t} p(\boldsymbol{x})^{\lambda_t}, \ \boldsymbol{S}_t(\boldsymbol{x}) := \nabla_{\boldsymbol{x}} \log p_t(\boldsymbol{x}) = (1 - \lambda_t)\boldsymbol{S}_{base}(\boldsymbol{x}) + \lambda_t \boldsymbol{S}(\boldsymbol{x}), \quad (6)$$

*where $\{\lambda_t\}_{t=0}^{T-1}$ is a non-negative decreasing sequence satisfying $\lambda_0 = 1$.*

Intuitively, we expect the distance between two neighboring distributions $p_t(\boldsymbol{x})$ and $p_{t+1}(\boldsymbol{x})$ to be not too large so that it would be easy to construct a conditional distribution $q_t(\boldsymbol{x}_t|\boldsymbol{x}_{t+1})$ such that $p_t(\boldsymbol{x}_t) \approx \int q_t(\boldsymbol{x}_t|\boldsymbol{x}_{t+1})p_{t+1}(\boldsymbol{x}_{t+1})\mathrm{d}\boldsymbol{x}_{t+1}$. Note that the auxiliary bridge $\{p_t(\boldsymbol{x})\}_{t=0}^{T-1}$ does not necessarily need to have analytical pdfs (up to a constant). In fact, it suffices if they have tractable score functions $\{\boldsymbol{S}_t(\boldsymbol{x})\}_{t=0}^{T-1}$ which lead to another type of auxiliary bridge (Example 2 in Section 4).

### 3.2 Sequential training of HSIVI

Given the auxiliary distributions $\{p_t(\boldsymbol{x}_t)\}_{t=0}^{T-1}$, a natural approach is to progressively train the hierarchical semi-implicit distribution $q_t(\boldsymbol{x}_t; \phi_{\geq t})$ to match $p_t(\boldsymbol{x}_t)$ from $t = T-1$ to $t = 0$. Let the parameters $\phi_t$ in the $t$-th conditional layer be independent across different $t$s. We first train $q_{T-1}(\boldsymbol{x}_{T-1}; \phi_{T-1})$ to match $p_{T-1}(\boldsymbol{x}_{T-1})$ by optimizing $\phi_{T-1}$ w.r.t. the single-layer SIVI objective $\mathcal{L}_{\text{SIVI-}f}\left(p_{T-1}(\boldsymbol{x}_{T-1})\|q_{T-1}(\boldsymbol{x}_{T-1}; \phi_{T-1})\right)$. For $t = T-2, \ldots, 0$, given the trained semi-implicit distribution $q_{t+1}(\boldsymbol{x}_{t+1}; \phi_{\geq t+1})$, we can fix it as the mixing layer and train the $t$-th conditional layer $q_t(\boldsymbol{x}_t|\boldsymbol{x}_{t+1}; \phi_t)$ by optimizing $\phi_t$ w.r.t. the single-layer SIVI objective $\mathcal{L}_{\text{SIVI-}f}\left(p_t(\boldsymbol{x}_t)\|q_t(\boldsymbol{x}_t; \phi_{\geq t})\right)$ as well. Note this is fine as the mixing layer can be implicit in SIVI. Here, $f$ is some distance criterion, e.g. $\mathcal{L}_{\text{SIVI-LB}}$ in equation (2) or $\mathcal{L}_{\text{SIVI-SM}}$ in equation (4). In this article, we mainly focus on $\mathcal{L}_{\text{SIVI-LB}}$ and $\mathcal{L}_{\text{SIVI-SM}}$, while other distance criteria can also be applied. We summarize this sequential training procedure in Algorithm 1.

**Score based training** In addition to the common assumption that $p_t(\boldsymbol{x})$ is known up to a constant, it is worth noting that $\mathcal{L}_{\text{SIVI-LB}}$ is also applicable when only the score functions $\{\boldsymbol{S}_t(\boldsymbol{x})\}_{t=0}^{T-1}$ are available which is important for the diffusion bridge construction of auxiliary distributions in Example 2. Concretely, assume $q_t(\boldsymbol{x}_t|\boldsymbol{x}_{t+1}; \phi_t)$ is induced by a parametrized transform $\boldsymbol{x}_t = \boldsymbol{h}_t(\boldsymbol{x}_{t+1}, \boldsymbol{\epsilon}; \phi_t)$

where $\boldsymbol{\epsilon} \sim p_{\boldsymbol{\epsilon}}(\boldsymbol{\epsilon})$ is a random noise. The only term in $\mathcal{L}_{\text{SIVI-LB}}\left(p_t(\boldsymbol{x}_t) \| q_t(\boldsymbol{x}_t; \phi_{\geq t})\right)$ containing $p_t(\boldsymbol{x}_t)$ is $\mathbb{E}_{q_t(\boldsymbol{x}_t; \phi_{\geq t})} \log p_t(\boldsymbol{x}_t)$ (see equation (2)) whose gradient takes the form

$$\nabla_{\phi_t} \mathbb{E}_{q_t(\boldsymbol{x}_t; \phi_{\geq t})} \log p_t(\boldsymbol{x}_t) = \mathbb{E}_{q_{t+1}(\boldsymbol{x}_{t+1}; \phi_{\geq t+1}) p_{\boldsymbol{\epsilon}}(\boldsymbol{\epsilon})} \boldsymbol{S}_t\left(\boldsymbol{h}_t(\boldsymbol{x}_{t+1}, \boldsymbol{\epsilon}; \phi_t)\right) \nabla_{\phi_t} \boldsymbol{h}_t(\boldsymbol{x}_{t+1}, \boldsymbol{\epsilon}; \phi_t). \quad (7)$$

In the training of HSIVI-SM, each term $\mathcal{L}_{\text{SIVI-SM}}\left(p_t(\boldsymbol{x}_t) \| q_t(\boldsymbol{x}_t; \phi_{\geq t})\right)$ involves a nested optimization of $\boldsymbol{f}_t(\boldsymbol{x}_t; \psi_t)$. When the score functions are computationally expensive, we find that an alternative parametrization $\boldsymbol{f}_t(\boldsymbol{x}_t; \psi_t) := \boldsymbol{S}_t(\boldsymbol{x}_t) - \boldsymbol{g}_t(\boldsymbol{x}_t; \psi_t)$ is useful to avoid the time-consuming evaluation of $\boldsymbol{S}_t(\boldsymbol{x}_t)$ when optimizing $\psi_t$ in equation (4). The reason for this lies in Proposition 1. See Appendix C.2 for the proof of Proposition 1.

**Proposition 1.** *Let* $q_t(\boldsymbol{x}_t, \boldsymbol{x}_{t+1}; \phi_{\geq t}) = q_t(\boldsymbol{x}_t | \boldsymbol{x}_{t+1}; \phi_t) q_{t+1}(\boldsymbol{x}_{t+1}; \phi_{\geq t+1})$. *The minimax optimization of* $\mathcal{L}_{\text{SIVI-SM}}\left(p_t(\boldsymbol{x}_t) \| q_t(\boldsymbol{x}_t; \phi_{\geq t})\right)$ *is equivalent to*

$$\min_{\phi_t} \quad \mathbb{E}_{q_t(\boldsymbol{x}_t, \boldsymbol{x}_{t+1}; \phi_{\geq t})} \left[\boldsymbol{S}_t(\boldsymbol{x}_t) - \boldsymbol{g}_t(\boldsymbol{x}_t; \psi_t)\right]^T \left[\boldsymbol{S}_t(\boldsymbol{x}_t) + \boldsymbol{g}_t(\boldsymbol{x}_t; \psi_t) - 2\nabla_{\boldsymbol{x}_t} \log q_t(\boldsymbol{x}_t | \boldsymbol{x}_{t+1}; \phi_t)\right],$$

$$\min_{\psi_t} \quad \mathbb{E}_{q_t(\boldsymbol{x}_t, \boldsymbol{x}_{t+1}; \phi_{\geq t})} \|\boldsymbol{g}_t(\boldsymbol{x}_t; \psi_t) - \nabla_{\boldsymbol{x}_t} \log q_t(\boldsymbol{x}_t | \boldsymbol{x}_{t+1}; \phi_t)\|_2^2.$$

**Marginal approximation v.s. joint approximation** Previous works (Bernton et al., 2019; Bao et al., 2022) often construct a joint distribution $p(\boldsymbol{x}_{0:T})$ and minimize $\text{KL}(p(\boldsymbol{x}_{0:T-1}) \| q(\boldsymbol{x}_{0:T-1}))$ where $q(\boldsymbol{x}_{0:T-1})$ is a variational distribution. In HSIVI, we directly approximate $p_t(\boldsymbol{x}_t)$ using the semi-implicit variational distributions. When $p(\boldsymbol{x}_{0:T-1})$ is complex and $T$ is small, the variational distribution $q(\boldsymbol{x}_{0:T-1})$ may be insufficient to fully capture the joint distribution $p(\boldsymbol{x}_{0:T-1})$. For example, the optimal fit of the joint distribution for diffusion models established by Analytic-DPM (Bao et al., 2022) does not guarantee that the marginal distributions would be approximated well (see Table 2 for comparison).

## 4 Application to diffusion model acceleration

### 4.1 Review of diffusion models

Recently, diffusion models have shown great success on many generative modeling benchmarks, including image generation (Ho et al., 2020; Song et al., 2020a,b), graph generation (Niu et al., 2020), and text generation (Austin et al., 2021). Diffusion models work by adding noise to the training data in the forward process and then removing the noise to recover the data in the backward process, which can be integrated into a general stochastic differential equation (SDE) framework. The forward process $\{\boldsymbol{u}_s\}_{s \in [0,L]}$ is usually described by

$$\mathrm{d}\boldsymbol{u}_s = \boldsymbol{f}(\boldsymbol{u}_s, s)\mathrm{d}s + g(s)\mathrm{d}\boldsymbol{w}_s, \quad \boldsymbol{u}_0 \sim p_0(\cdot), \quad (8)$$

where $p_0(\cdot)$ is the data distribution, $\boldsymbol{w}_s$ is a standard Brownian motion, and $\boldsymbol{f}(\boldsymbol{u}_s, s)$ and $g(s)$ are the drift and diffusion coefficients respectively. To generate samples from the data distribution, one can run the following backward process

$$\mathrm{d}\boldsymbol{u}_s = [\boldsymbol{f}(\boldsymbol{u}_s, s) - g^2(s)\nabla_{\boldsymbol{u}_s} \log p_s(\boldsymbol{u}_s)]\mathrm{d}s + g(s)\mathrm{d}\bar{\boldsymbol{w}}_s, \quad \boldsymbol{u}_L \sim p_L(\cdot), \quad (9)$$

where $p_s(\cdot)$ is the pdf of $\boldsymbol{u}_s$ and $\bar{\boldsymbol{w}}_s$ is a standard Brownian motion when time flows from $L$ to $0$. As the score function $\nabla_{\boldsymbol{u}_s} \log p_s(\boldsymbol{u}_s)$ is intractable, we need to estimate it by denoising score matching (Vincent, 2011; Song et al., 2020b). See more details of diffusion models and the training objectives in Appendix A.

### 4.2 Diffusion model acceleration via HSIVI

While diffusion models prove effective for generative modeling, it often takes a large number of discretization steps in the backward process (9) to produce high quality samples, which caps their potential for real time applications. Note that the forward process (8) naturally provides another type of auxiliary bridge, which combined with HSIVI, can be used to accelerate the sampling process of diffusion models.

**Example 2** (Diffusion Bridge). *Consider the forward process* $\{\boldsymbol{u}_s\}_{s \in [0,L]}$ *with* $L > 0$ *(defined in equation (8)) in diffusion models. We choose $T$ discrete time steps* $0 \approx s_0 < \cdots < s_{T-1} \leq L$ *and*

let $\boldsymbol{x}_t := \boldsymbol{u}_{s_t}$ with probability density function $p_t(\cdot)$. Assume each auxiliary distributions $p_t(\cdot)$ for $0 \le t \le T - 1$ admits a score function as

$$\boldsymbol{S}_t(\boldsymbol{x}) := \nabla_{\boldsymbol{x}} \log p_t(\boldsymbol{x}) \approx \boldsymbol{S}^*(\boldsymbol{x}, s_t),\ 0 \le t \le T - 1,$$

where $\boldsymbol{S}^*(\boldsymbol{x}, s)$ is a pre-trained score model with the denoising score matching loss (equation (13) in Appendix A). Let us denote $\boldsymbol{S}^*(\boldsymbol{x}, s_t)$ by $\boldsymbol{S}_t^*(\boldsymbol{x})$ for short. With sufficient samples from the data distribution $p_0(\boldsymbol{x})$ and model capacity, the approximation $\boldsymbol{S}_t^*(\boldsymbol{x})$ can be reasonably accurate for almost all $\boldsymbol{x}$ and $t$ (Song et al., 2020b).

As the pre-trained score model provides a diffusion bridge from the simple distribution $p_{T-1}$ (e.g., standard Gaussian) to the data distribution, we can train the hierarchical semi-implicit distributions to approximate the diffusion bridge within the HSIVI framework. Given the expressiveness of hierarchical semi-implicit distributions, we may expect an accurate approximation of the data distribution with a small number $T$ and hence acceleration can be achieved.

However, the memory usage during the sequential training process for HSIVI might be large because of the necessity for independent parameters. Therefore, we may employ a parameter sharing scheme which is commonly assumed in diffusion models (Song & Ermon, 2019; Ho et al., 2020) such that different conditional layers share the same parameters $\phi$. Note that sequential training is not suitable in this setting. Therefore, we propose a joint training procedure that minimizes a weighted sum of the SIVI objectives

$$\mathcal{L}_{\text{HSIVI-}f}(\phi) = \sum_{t=0}^{T-1} \beta(t) \mathcal{L}_{\text{SIVI-}f}\left(p_t(\boldsymbol{x}_t) \| q_t(\boldsymbol{x}_t; \phi)\right), \tag{10}$$

where $\beta(t) : \{0, \dots, T - 1\} \to \mathbb{R}_+$ is a positive weighting function and $f$ is some distance criterion. See Algorithm 2 in Appendix C.3 for more details of joint training.

More specifically, in this work, we mainly focus on building the diffusion bridge with variance preserving SDE (VP-SDE) (Song et al., 2020b) such that $\boldsymbol{u}_s | \boldsymbol{u}_0 \sim \mathcal{N}(\sqrt{\alpha(s)} \boldsymbol{u}_0, (1 - \alpha(s))\mathbf{I})$ with a decreasing function $\alpha(s)$ of $s$. We use $\mathcal{L}_{\text{HSIVI-SM}}$ in equation (10) for training and set the weighting function $\beta(t) = 1 - \alpha(s_t)$ as recommended in Song et al. (2020b), which tends to train layers that are far from $t = 0$ first during the training, resembling the sequential training. Another popular formulation of diffusion models is to fit a noise model $\boldsymbol{\epsilon}^*(\boldsymbol{x}, s)$ that predicts the noise added to a noisy sample $\boldsymbol{x}$ at time $s$ (Ho et al., 2020). HSIVI-SM also generalizes to the case where a pre-trained noise model is available. The pre-trained noise model forms a (generalized) diffusion bridge by letting $\boldsymbol{\epsilon}_t^*(\boldsymbol{x}) = \boldsymbol{\epsilon}^*(\boldsymbol{x}, s_t)$, and we call the corresponding training method "$\epsilon$-training". We provide a reparametrized objective function $\tilde{\mathcal{L}}_{\text{HSIVI-SM}}$ for $\epsilon$-training in Appendix C.4.

Several efforts have been made to accelerate the sampling process of diffusion models, including faster numerical ordinary differential equation (ODE) solvers (Song et al., 2020a; Zhang & Chen, 2022; Lu et al., 2022) and distillation techniques (Luhman & Luhman, 2021; Salimans & Ho, 2022; Zheng et al., 2022). Our approach is different from these previous efforts in that we accelerate the stochastic diffusion model directly (hence would provide more diverse samples (Figure 6)) and do not require sampling datasets from the diffusion models prior to distillation which is computationally expensive. From a Bayesian perspective, HSIVI is related to Song & Ermon (2019), where the authors used the annealed Langevin dynamics guided by a pre-trained score model to sample from the data distribution. By solving this problem using a variational inference approach, HSIVI enjoys faster sampling speed and scales better to high-dimensional data.

## 5   Experiments

In this section, we first compare HSIVI to its single-layer counterpart, SIVI, on two inference tasks. We use the sequential training method where each conditional layer in the hierarchical semi-implicit variational distributions has independent parameters. We then apply HSIVI-SM to diffusion model acceleration on various datasets. As the memory consumption for generative models is large, we use the joint training method where the conditional layers in hierarchical semi-implicit distributions have shared parameters across different $t$s. For all experiments, each conditional layer is modeled as a Gaussian distribution with parametrized mean and variance. More details of the model architectures and hyper-parameters are included in Appendix E. The code is available at `https://github.com/longinYu/HSIVI`.

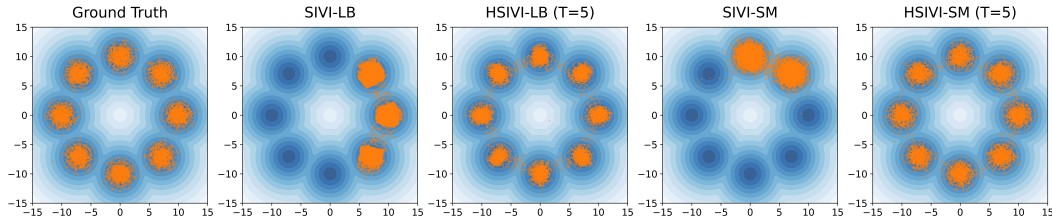

Figure 2: Comparison of 10,000 generated samples from SIVI and 5-layer HSIVI on a two-dimensional Gaussian mixture model (blue).

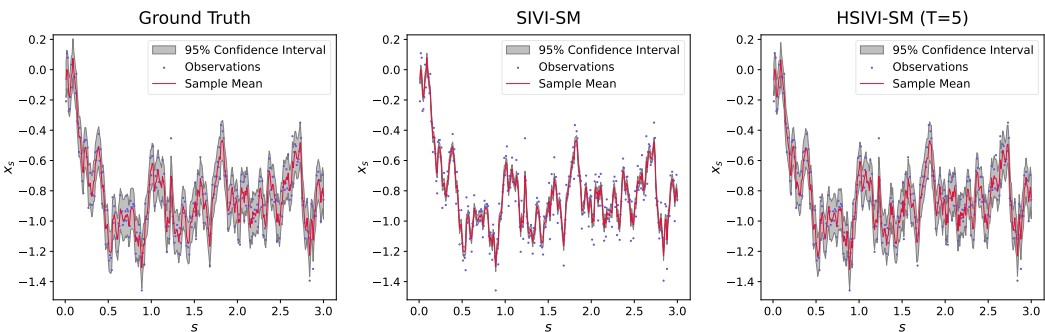

Figure 3: The posterior estimates obtained by different methods. For each method, we collect 100,000 samples to calculate the sample mean and confidence interval.

## 5.1 Target distribution approximation

**Gaussian mixture model**   We first evaluate HSIVI and SIVI on a two-dimensional Gaussian mixture model. The target distribution $p(\boldsymbol{x})$ takes the form $p(\boldsymbol{x}) = \sum_{i=1}^{8} 1/8 \cdot \mathcal{N}(\boldsymbol{x}; \boldsymbol{\mu}_i, \sigma^2 \mathbf{I})$ where $\boldsymbol{\mu}_i = [10\cos(\frac{i\pi}{4}), 10\sin(\frac{i\pi}{4})]^T$, $\sigma = 1$. For HSIVI, we construct an auxiliary bridge of $T = 5$ with geometric interpolation in Example 1, where $p_{\text{base}}(\boldsymbol{x}) = \mathcal{N}(\boldsymbol{x}; \mathbf{0}, \mathbf{I})$ and $\lambda_t = 1 - t/5$. The results are presented in Figure 2. Note that the modes in this Gaussian mixture model are far apart from each other, and both SIVI-LB and SIVI-SM are trapped in local modes. In contrast, both HSIVI-LB and HSIVI-SM discover all modes and provide an accurate approximation of the target distribution with HSIVI-SM being better for recovering the right scale of variance.

**High-dimensional conditioned diffusion**   The second example is a high-dimensional Bayesian inference problem arising from the following Langevin SDE

$$\mathrm{d}x_s = 10x_s(1 - x_s^2)\mathrm{d}s + \mathrm{d}w_s, \tag{11}$$

where $x_0 = 0$ and $w_s$ is a one-dimensional standard Brownian motion. This system describes the motion of a particle with negligible mass trapped in an energy potential with thermal fluctuations represented by the Brownian forcing (Cui et al., 2016). Using an Euler-Maruyama scheme with step size $\Delta s = 0.01$ on a time interval $[0, 3]$, we discretize the SDE (11) into $\boldsymbol{x} = (x_{d_1}, \ldots, x_{d_{300}})$ where $d_i = 0.01i$, which gives the prior distribution $p_{\text{prior}}(\boldsymbol{x})$ of the 300-dimensional variable $\boldsymbol{x}$. The noisy observations $\boldsymbol{y}$ is obtained by $\boldsymbol{y} = \boldsymbol{x} + \boldsymbol{\xi}$, where $\boldsymbol{\xi} \sim \mathcal{N}(\mathbf{0}, \sigma^2 \mathbf{I})$ with $\sigma = 0.1$. Our goal is to infer the posterior distribution of the latent states $p(\boldsymbol{x}|\boldsymbol{y}) \propto p_{\text{prior}}(\boldsymbol{x})p(\boldsymbol{y}|\boldsymbol{x})$. The ground truth is formed by running 100,000 independent stochastic gradient Langevin dynamics (SGLD) chains with a step size of 0.0001 and collecting the results after 10,000 iterations.

For HSIVI, we form the auxiliary bridge using geometric interpolation with $p_{\text{base}}(\boldsymbol{x}) = \mathcal{N}(\boldsymbol{x}; \boldsymbol{y}, \sigma^2 \mathbf{I})$ and $\lambda_t = 1 - t/(T - 1)$ for $t = 0, \ldots, T - 1$. Figure 3 shows the estimated posteriors obtained by different methods. We see that SIVI-SM severely underestimates the variance. With $T = 5$ layers, HSIVI-SM fits the variance better and hence provides more accurate posterior estimates. For both HSIVI-SM and HSIVI-LB, the estimated covariance matrix becomes more accurate as $T$ increases (Table 4 in Appendix D.2), demonstrating the effectiveness of hierarchical models for fitting complicated distributions.

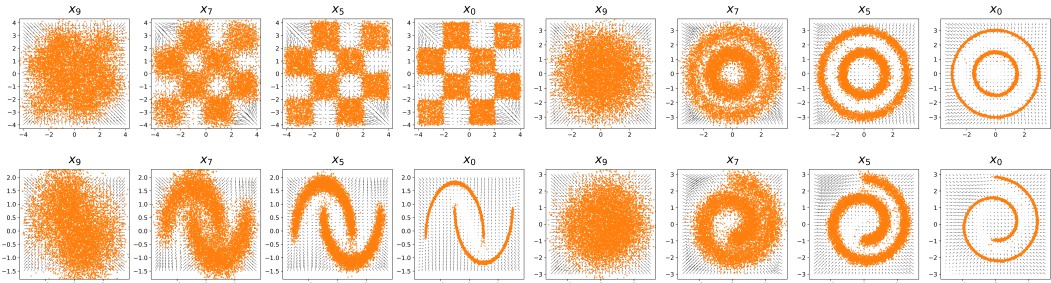

Figure 4: Sample trajectories generated from 10-layer HSIVI-SM on four 2D toy examples. The arrows represent the estimated score function in HSIVI-SM. The sample size is 10,000.

Table 1: JS divergences between the target distribution and the variational approximation on the four toy datasets. The results of HSIVI-SM are averaged by 5 independent runs with standard deviation in the subscripts. JS divergences are calculated by the ITE package (Szabó, 2014) with 10,000 samples.

| Name | $T = 5$ | | | $T = 10$ | | | $T = 1000$ |
|------|------|------|----------|------|------|----------|------|
| | DDPM | DDIM | HSIVI-SM | DDPM | DDIM | HSIVI-SM | DDPM |
| Checkerboard | 0.891 | 0.591 | $\mathbf{0.068}_{\pm 0.006}$ | 0.521 | 0.373 | $\mathbf{0.030}_{\pm 0.005}$ | 0.058 |
| Swissroll | 1.037 | 0.332 | $\mathbf{0.126}_{\pm 0.006}$ | 0.334 | 0.164 | $\mathbf{0.082}_{\pm 0.003}$ | 0.042 |
| Circles | 0.907 | 0.397 | $\mathbf{0.083}_{\pm 0.015}$ | 0.364 | 0.201 | $\mathbf{0.073}_{\pm 0.005}$ | 0.032 |
| Moons | 0.961 | 0.355 | $\mathbf{0.096}_{\pm 0.013}$ | 0.352 | 0.137 | $\mathbf{0.059}_{\pm 0.007}$ | 0.036 |

## 5.2 Diffusion model acceleration

**2D toy examples** In this toy model example, we test four synthetic 2D datasets: Checkerboard, Circles, Moons, and Swissroll (Pedregosa et al., 2011). We first pre-train the score model $\boldsymbol{S}^*(\boldsymbol{x}, s)$ for $s \in [0, 1]$ with quadratic noise schedule $1 - \alpha(s) = s^2$. For constructing the $T$-layer diffusion bridge, we select $\{s_t\}_{t=0}^{T-1}$ so that $1 - \alpha(s_t) = [0.01 + (\sqrt{0.8} - 0.01)t/T]^2$. Figure 4 shows the sample trajectories ($\boldsymbol{x}_9$, $\boldsymbol{x}_7$, $\boldsymbol{x}_5$ and $\boldsymbol{x}_0$) progressively generated from 10-layer HSIVI-SM. We see clearly how the semi-implicit distributions are guided towards the target distribution and all modes are discovered. We also report the Jensen-Shannon (JS) divergence between the target distributions and the estimated distributions in Table 1. We see that HSIVI-SM significantly improves upon DDIM and DDPM in both cases with 5 and 10 steps. Also, 10-layer HSIVI-SM is comparable to DDPM with 1000 full steps. See Figure 10 in Appendix D.3 for visualization of samples from different methods.

**MNIST** On MNIST, we use the noise model $\boldsymbol{\epsilon}^*(\boldsymbol{x}, s)$ instead of the score model and use $\epsilon$-training to train HSIVI-SM. The structure of $\boldsymbol{\epsilon}^*(\boldsymbol{x}, s)$ follows the UNet in Ho et al. (2020) by reducing the number of input and output channels to one. With the same noise schedule employed in Song et al. (2020a), we first pre-train the noise model $\boldsymbol{\epsilon}^*(\boldsymbol{x}, s)$ with 1000 discretization steps and then form the $T$-layer diffusion bridge for HSIVI-SM by selecting $T$ discrete time steps. Figure 5 shows the samples from DDPM, DDIM, and HSIVI-SM with $T = 5$ steps. We see that the samples produced by HSIVI-SM are much cleaner and more recognizable than those produced by DDPM and DDIM.

**CIFAR-10, CelebA & ImageNet** On both CIFAR-10 and CelebA, the structure of our pre-trained noise model $\boldsymbol{\epsilon}^*(\boldsymbol{x}, s)$ follows the UNet structure(Ronneberger et al., 2015) employed by Ho et al. (2020), instead of the huge VP deep continuous-time model (Song et al., 2020b) that has more channels and layers. We also provide additional results on ImageNet (64×64) with more powerful pre-trained score nets in (Nichol & Dhariwal, 2021)(bigger models with more parameters). Since this generative modeling has been formulated as a score-based VI problem, we do not have to use any training data for training HSIVI-SM.

Following the noise schedule employed in Song et al. (2020a), we first pre-train the noise model $\boldsymbol{\epsilon}^*(\boldsymbol{x}, s)$ with 1000 discretization steps and then form the $T$-layer diffusion bridge for HSIVI-SM

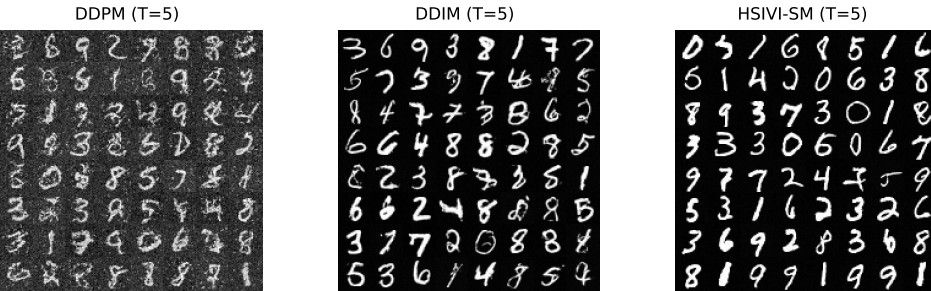

Figure 5: Comparison of the quality of uncurated samples generated by DDPM, DDIM, and HSIVI-SM with 5 discrete time steps on MNIST.

Table 2: Sample quality measured by FID ($\downarrow$) on CIFAR-10, CelebA and ImageNet with a varying number of function evaluations (NFE). Results of baselines are calculated by running their official codes, where the architectures of score model (or noise model) are the UNet employed in Ho et al. (2020) for CIFAR-10 and CelebA and (Nichol & Dhariwal, 2021) in ImageNet.

| Dataset | CIFAR-10 (32×32) | | | CelebA (64×64) | | | ImageNet (64×64) | | |
|---|---|---|---|---|---|---|---|---|---|
| NFE | 5 | 10 | 15 | 5 | 10 | 15 | 5 | 10 | 15 |
| DDPM (Ho et al., 2020) | 320.16 | 278.65 | 198.00 | 366.10 | 309.95 | 206.92 | 402.68 | 358.80 | 284.00 |
| DDIM (Song et al., 2020a) | 41.53 | 13.73 | 8.78 | 27.38 | 10.89 | 7.78 | 147.03 | 42.31 | 24.85 |
| FastDPM (Kong & Ping, 2021) | 67.64 | 9.85 | 6.16 | 27.63 | 15.44 | 12.05 | N/A | N/A | N/A |
| Analytic-DDPM (Bao et al., 2022) | 93.16 | 34.54 | 20.03 | 50.92 | 28.93 | 21.84 | N/A | 60.65 | 45.98 |
| Analytic-DDIM (Bao et al., 2022) | 51.86 | 14.08 | 8.65 | 29.40 | 15.74 | 12.25 | N/A | 70.62 | 41.56 |
| DPM-Solver-fast (Lu et al., 2022) | 329.13 | 10.89 | 4.67 | 355.96 | 6.76 | 2.98 | 402.43 | 28.96 | 20.03 |
| **HSIVI-SM (ours)** | **6.27** | **4.31** | **4.17** | **6.22** | **3.09** | **2.23** | **40.43** | **17.67** | **15.49** |

by selecting $T$ discrete time steps as before. For HSIVI-SM with $\epsilon$-training, the conditional layer $q_t(\cdot|\boldsymbol{x}_{t+1}; \phi)$ is modeled as a Gaussian distribution with mean $\boldsymbol{\mu}_t(\boldsymbol{x}_{t+1}; \phi^\mu)$ and diagonal variance matrix $\boldsymbol{\Sigma}_t(\phi^\sigma)$ where $\{\phi^\mu, \phi^\sigma\} = \phi$ are the variational parameters. In our implementations, both $\boldsymbol{\mu}_t(\boldsymbol{x}_{t+1}; \phi^\mu)$ and $\boldsymbol{f}_t(\boldsymbol{x}_t; \psi)$ use the same architecture as $\boldsymbol{\epsilon}^*(\boldsymbol{x}, s)$. The number of layers, which is also the number of function evaluations (NFE), is set to be $T = 5, 10, 15$ in our experiments. We train HSIVI-SM with the same setting for $T = 10, 15$. The 5-layer HSIVI-SM is trained by further fine-tuning the well-trained 15-layer HSIVI-SM and we find this strategy leads to better results. During each nested training loop of $\boldsymbol{f}_t(\boldsymbol{x}_t; \psi)$, we update $\psi$ 20 times before each update of $\phi$, since we find $\boldsymbol{f}_t(\boldsymbol{x}_t; \psi)$ needs more training empirically to provide reliable guidance.

For each method, we draw 50,000 samples and use the Fréchet inception distance (FID) score (Karras et al., 2022) to evaluate the sample quality (Table 2). We find that HSIVI-SM performs on par or better than the other baselines on both CIFAR-10 and CelebA, and the advantage is evident when the NFE is small. The sampling trajectories of 10-layer HSIVI-SM on CelebA with the same starting point but different random seeds are shown in Figure 6. We see that HSIVI-SM is capable of producing more diverse samples due to its stochastic nature, which is different from existing ODE based fast diffusion model samplers.



Figure 6: Sample trajectories of 10-layer HSIVI-SM with the same starting point $\boldsymbol{x}_{10}$ on CelebA.

### 5.3 Additional Study

**Ablation of layers number** In Figure 7, We provide a failure case on fitting the checkerboard target with diffusion bridge, demonstrating that the HSIVI-SM algorithm fails when the layer number $T$ is small (the distances of auxiliary distributions at successive time steps are large) on a checkerboard

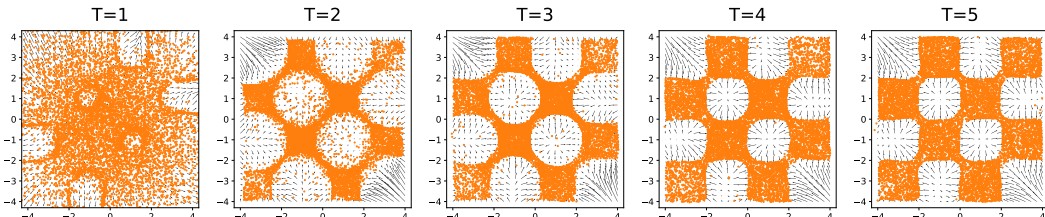

Figure 7: Failure cases of HSIVI-SM. The quivers show the estimated score by the $f$ function. $T$ is the layers number of HSIVI-SM. The generated samples in orange show that smaller $T$ may fail on this example.

Table 3: Comparison of non-isotropic conditional layers and isotropic conditional layers on CIFAR10, the sample quality is measured by FID ($\downarrow$).

| NFE | DDPM | DDIM | HSIVI-SM (isotropic) | HSIVI-SM (non-isotropic) |
|-----|------|------|----------------------|--------------------------|
| 5 | 320.16 | 41.53 | 7.33 | **6.27** |
| 10 | 278.65 | 13.73 | 4.78 | **4.31** |
| 15 | 198.00 | 8.78 | 4.46 | **4.17** |

distribution. In fact, the score function on the checkerboard target is sharp on the boundaries but vanishes elsewhere. Therefore, fitting this target distribution is somewhat challenging.

**Ablation of the variational family**   To validate the improvement of HSIVI-SM on diffusion models, we train HSIVI-SM with isotropic conditional layers in consistency with denoising-diffusion sampling, like DDPM and DDIM. We report the results of FID on the CIFAR-10 dataset in Table 3. These results provide further evidence for the statement outlined in Section 3.2. HSIVI-SM matches the marginal distributions $q_t(\boldsymbol{x}_t)$ and $p_t(\boldsymbol{x}_t)$ directly via score matching and would ensure a better fit for $p_0(\boldsymbol{x}_0)$. The enhancement of HSIVI-SM over DDPM stems not only from its more expressive variational distribution but also from the direct alignment of the marginal distributions.

# 6   Conclusions

We introduced HSIVI, a hierarchical semi-implicit variational inference method that enables more expressive multi-layer construction of semi-implicit distributions. Given appropriate auxiliary distributions that interpolate between a simple base distribution and the target distribution, the conditional layers in hierarchical semi-implicit distributions can be progressively trained one layer after another. In experiments, we showed that HSIVI outperforms previous single-layer SIVI methods on several Bayesian inference tasks with complicated posteriors. HSIVI can also be used to accelerate the sampling process of diffusion models, where pre-trained score networks serve as a natural sequence of bridging distributions, which allows for direct acceleration of the stochastic diffusion model and does not require expensive sampling from the diffusion models during training. We showed that HSIVI can produce high quality samples comparable to or better than existing fast diffusion model samplers with few function evaluations on various datasets. Limitations are discussed in Appendix F.

# Acknowledgements

This work was supported by National Natural Science Foundation of China (grant no. 12201014 and grant no. 12292983). The research of Cheng Zhang was supported in part by National Engineering Laboratory for Big Data Analysis and Applications, the Key Laboratory of Mathematics and Its Applications (LMAM) and the Key Laboratory of Mathematical Economics and Quantitative Finance (LMEQF) of Peking University. The authors appreciate the anonymous NeurIPS reviewers for their constructive feedback.

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

## A  Details of diffusion models

Diffusion models work by adding noise to the training data in the forward process and then removing the noise to recover the data in the backward process, which can be integrated into a general stochastic differential equation (SDE) framework (Song et al., 2020b). The forward process $\{\boldsymbol{u}_s\}_{s\in[0,L]}$ is usually described by the SDE

$$\mathrm{d}\boldsymbol{u}_s = \boldsymbol{f}(\boldsymbol{u}_s, s)\mathrm{d}s + g(s)\mathrm{d}\boldsymbol{w}_s, \quad \boldsymbol{u}_0 \sim p_0(\cdot),$$

where $p_0(\cdot)$ is the data distribution, $\boldsymbol{w}_s$ is a standard Brownian motion, $\boldsymbol{f}(\boldsymbol{u}_s, s)$ and $g(s)$ are the drift and diffusion coefficient respectively. To generate samples from the data distribution, one can run the following reversed SDE

$$\mathrm{d}\boldsymbol{u}_s = [\boldsymbol{f}(\boldsymbol{u}_s, s) - g^2(s)\nabla_{\boldsymbol{u}_s}\log p_s(\boldsymbol{u}_s)]\mathrm{d}s + g(s)\mathrm{d}\bar{\boldsymbol{w}}_s, \quad \boldsymbol{u}_L \sim p_L(\cdot),$$

where $p_s(\cdot)$ is the probability density function (pdf) of $\boldsymbol{u}_s$ and $\bar{\boldsymbol{w}}_s$ is a standard Brownian motion when time flows from $L$ to $0$. There exists deterministic process shares the same marginal probability densities $\{p_s(\cdot)\}_{s\in[0,L]}$ described by the following ordinary differential equation (ODE)

$$\mathrm{d}\boldsymbol{u}_s = [\boldsymbol{f}(\boldsymbol{u}_s, s) - \frac{1}{2}g^2(s)\nabla_{\boldsymbol{u}_s}\log p_s(\boldsymbol{u}_s)]\mathrm{d}s, \quad \boldsymbol{u}_L \sim p_L(\cdot),$$

called probability flow (PF) ODE.

In practice, Song et al. (2020b) and Kingma et al. (2021) designed several examples of the forward process such that it diffuses the data distribution $p_0(\cdot)$ to a fixed unstructured distribution $p_L(\cdot)$. Here we mainly consider the Variance Preserving SDE (VP-SDE) used in DDPM (Ho et al., 2020; Song et al., 2020b). Let the drift coefficient $\boldsymbol{f}(\boldsymbol{u}_s, s) = \frac{\mathrm{d}\log\alpha(s)}{2\mathrm{d}s}\boldsymbol{u}_s$ and the diffusion coefficient $g^2(s) = -\frac{\mathrm{d}\log\alpha(s)}{\mathrm{d}s}$, where $\alpha(s) \in \mathbb{R}^+$ is a decreasing smooth function with $\alpha(0) = 1, \alpha(L) \approx 0$. Then the distribution of $\boldsymbol{u}_s$ conditioned on $\boldsymbol{u}_0$ is explicit as

$$\boldsymbol{u}_s|\boldsymbol{u}_0 \sim \mathcal{N}\left(\sqrt{\alpha(s)}\bar{\boldsymbol{x}}_0, (1-\alpha(s))\mathbf{I}\right), \text{ i.e. } \boldsymbol{u}_s = \sqrt{\alpha(s)}\boldsymbol{u}_0 + \sqrt{1-\alpha(s)}\boldsymbol{\epsilon}, \qquad (12)$$

where $\boldsymbol{\epsilon}$ is a standard Gaussian noise. In practice, diffusion models use a neural network $\boldsymbol{S}_\theta(\boldsymbol{u}_s, s)$ to approximate the score function $\boldsymbol{S}_\theta(\boldsymbol{u}_s, s)$ by optimizing the denoising score matching objective (Vincent, 2011)

$$\mathcal{L}_{\mathrm{dsm}}(\theta, \omega(s)) := \frac{1}{2}\int_0^L \omega(s)\mathbb{E}_{\boldsymbol{u}_0\sim p_0(\boldsymbol{u}_0), \boldsymbol{\epsilon}\sim\mathcal{N}(0,\mathbf{I})}\left\|\boldsymbol{S}_\theta(\boldsymbol{u}_s, s) + \boldsymbol{\epsilon}/\sqrt{1-\alpha(s)}\right\|_2^2\mathrm{d}s, \qquad (13)$$

where $\omega(s)$ is a positive weighting function. Instead of modeling the score function, Ho et al. (2020) proposed to predict the conditional noise $\boldsymbol{\epsilon}$ based on $\boldsymbol{u}_t$. This leads to the following DDPM loss

$$\mathcal{L}_{\mathrm{ddpm}}(\theta, \bar{\omega}(s)) := \frac{1}{2}\int_0^L \bar{\omega}(s)\mathbb{E}_{\boldsymbol{u}_0\sim p_0(\boldsymbol{u}_0), \boldsymbol{\epsilon}\sim\mathcal{N}(0,\mathbf{I})}\|\boldsymbol{\epsilon}_\theta(\boldsymbol{u}_s, s) - \boldsymbol{\epsilon}\|_2^2\mathrm{d}s, \qquad (14)$$

where $\bar{\omega}(s)$ is a positive weighting function. In fact, we have the relationship

$$\boldsymbol{S}_\theta(\boldsymbol{u}_s, s) = -\boldsymbol{\epsilon}_\theta(\boldsymbol{u}_s, s)/\sqrt{1-\alpha(s)}. \qquad (15)$$

We call $\mathcal{L}_{\mathrm{dsm}}$ "score-prediction" training and $\mathcal{L}_{\mathrm{ddpm}}$ "$\epsilon$-prediction" training.

With the pre-trained score model $\boldsymbol{S}_\theta(\boldsymbol{u}_s, s)$ or noise model $\boldsymbol{\epsilon}_\theta(\boldsymbol{u}_s, s)$, Song et al. (2020b) shows that the samples of $p_0(\cdot)$ can be generated by simulating the backward SDE, e.g. the sampling scheme of DDPM (Ho et al., 2020). Moreover, Bao et al. (2022) proposed Analytic-DPM, the optimal discretization form responding to the KL divergence of the joint distribution on the discrete time steps. Also, several high-order ODE solvers (Song et al., 2020a; Zhang & Chen, 2022; Lu et al., 2022) were proposed to achieve faster sampling.

## B  More details of UIVI

Unlike optimizing the surrogate ELBO, Titsias & Ruiz (2019) proposed unbiased implicit variational inference (UIVI) which relies on an unbiased gradient estimator for the exact ELBO. To elaborate further, reparametrize the conditional $q_\phi(\boldsymbol{x}|\boldsymbol{z})$ such as $\boldsymbol{x} = T_\phi(\boldsymbol{z}, \boldsymbol{\epsilon}), \boldsymbol{\epsilon} \sim q_{\boldsymbol{\epsilon}}(\boldsymbol{\epsilon})$, then

$$\nabla_\phi \mathrm{ELBO} = \nabla_\phi \mathbb{E}_{\boldsymbol{\epsilon}\sim q_{\boldsymbol{\epsilon}}(\boldsymbol{\epsilon}), \boldsymbol{z}\sim q(\boldsymbol{z})}\left[\log p(D, \boldsymbol{x}) - \log q_\phi(\boldsymbol{x})|_{\boldsymbol{x}=T_\phi(\boldsymbol{z}, \boldsymbol{\epsilon})}\right]$$

$$= \mathbb{E}_{\boldsymbol{\epsilon}\sim q_{\boldsymbol{\epsilon}}(\boldsymbol{\epsilon}), \boldsymbol{z}\sim q(\boldsymbol{z})}\left[g_\phi^{\mathrm{mod}}(\boldsymbol{z}, \boldsymbol{\epsilon}) + g_\phi^{\mathrm{ent}}(\boldsymbol{z}, \boldsymbol{\epsilon})\right],$$

where

$$g_\phi^{\text{mod}}(z, \epsilon) := \nabla_x \log p(D, x)|_{x=T_\phi(z,\epsilon)} \nabla_\phi T_\phi(z, \epsilon),$$

$$g_\phi^{\text{ent}}(z, \epsilon) := -\mathbb{E}_{q_\phi(z'|x)} \nabla_x \log q_\phi(x|z')\big|_{x=T_\phi(z,\epsilon)} \nabla_\phi T_\phi(z, \epsilon). \tag{16}$$

The second gradient term $g_\phi^{\text{ent}}$ involves an expectation w.r.t. the reverse conditional $q_\phi(z'|x)$ which is estimated by an MCMC sampler in UIVI. However, the inner-loop MCMC runs may require long iterations for convergence.

## C    More details of HSIVI

### C.1    Score-based training of HSIVI-LB

In the sequential training of HSIVI-LB, although the objective $\mathcal{L}_{\text{SIVI-LB}}\left(p_t(x_t)\|q_t(x_t; \phi_{\geq t})\right)$ is calculated based on $p_t(x)$, the gradient of it w.r.t. $\phi_t$ has a closed form containing only the score function $S_t(x)$ without knowing the corresponding pdfs. This derivation is important in the tasks where score functions of the auxiliary distributions are tractable while pdfs (up to a constant) of them are unavailable (for example, the diffusion bridge in Example 2). Concretely, assume the $t$-th conditional layer $q_t(x_t|x_{t+1}; \phi_t)$ is induced by a parametrized transform $x_t = h_t(x_{t+1}, \epsilon; \phi_t)$ where $\epsilon \sim p_\epsilon(\epsilon)$ is a random noise, since $q_t(x_t|x_{t+1}; \phi_t)$ is reparametrizable according to Definition 1. The only term in $\mathcal{L}_{\text{SIVI-LB}}\left(p_t(x_t)\|q_t(x_t; \phi_{\geq t})\right)$ containing $p_t(x_t)$ is $\mathbb{E}_{q_t(x_t; \phi_{\geq t})} \log p_t(x_t)$ (see equation (2)) whose gradient takes the form

$$\nabla_{\phi_t} \mathbb{E}_{q_t(x_t; \phi_{\geq t})} \log p_t(x_t) = \nabla_{\phi_t} \mathbb{E}_{q_{t+1}(x_{t+1}; \phi_{\geq t+1}) p_\epsilon(\epsilon)} \log p_t(h_t(x_{t+1}, \epsilon; \phi_t))$$

$$= \mathbb{E}_{q_{t+1}(x_{t+1}; \phi_{\geq t+1}) p_\epsilon(\epsilon)} S_t\left(h_t(x_{t+1}, \epsilon; \phi_t)\right) \nabla_{\phi_t} h_t(x_{t+1}, \epsilon; \phi_t)$$

by the chain rule, where $\nabla_{\phi_t} h_t(x_{t+1}, \epsilon; \phi_t)$ is the jacobian matrix of $h_t(x_{t+1}, \epsilon; \phi_t)$.

In our implementation of HSIVI (in both sequential training and joint training), we generally assume the conditional layer $q_t(\cdot|x_{t+1}; \phi_t)$ is induced by

$$h_t(x_{t+1}, \epsilon; \phi_t) = \mu_t(x_{t+1}; \phi_t) + \Sigma_t^{1/2}(x_{t+1}; \phi_t)\epsilon \tag{17}$$

where $\Sigma_t(x_{t+1}; \phi_t)$ is a positive definite covariance matrix and $\epsilon \sim \mathcal{N}(0, I)$ is a standard multivariate gaussian variable. In equation (17), $\phi_t$ should be replaced by $\phi$ in the joint training case.

### C.2    Proof of Proposition 1

**Proposition 1.** *Let* $q_t(x_t, x_{t+1}; \phi_{\geq t}) = q_t(x_t|x_{t+1}; \phi_t)q_{t+1}(x_{t+1}; \phi_{\geq t+1})$. *The minimax optimization of* $\mathcal{L}_{\text{SIVI-SM}}\left(p_t(x_t)\|q_t(x_t; \phi_{\geq t})\right)$ *is equivalent to*

$$\min_{\phi_t} \quad \mathbb{E}_{q_t(x_t, x_{t+1}; \phi_{\geq t})} \left[S_t(x_t) - g_t(x_t; \psi_t)\right]^T \left[S_t(x_t) + g_t(x_t; \psi_t) - 2\nabla_{x_t} \log q_t(x_t|x_{t+1}; \phi_t)\right],$$

$$\min_{\psi_t} \quad \mathbb{E}_{q_t(x_t, x_{t+1}; \phi_{\geq t})} \|g_t(x_t; \psi_t) - \nabla_{x_t} \log q_t(x_t|x_{t+1}; \phi_t)\|_2^2.$$

**Proof of Propsition 1**    The minimax optimization problem of $\mathcal{L}_{\text{SIVI-SM}}\left(p_t(x_t)\|q_t(x_t; \phi_{\geq t})\right)$ is

$$\min_{\phi_t} \max_{\psi_t} \mathbb{E}_{q_t(x_t, x_t; \phi_{\geq t})} \left[2f_t(x_t; \psi_t)^T[S_t(x_t) - \nabla_{x_t} \log q_t(x_t|x_{t+1}; \phi_t)] - \|f_t(x; \psi_t)\|_2^2\right]$$

according to equation (4). For minimization w.r.t. $\phi_t$, this target is equivalent to

$$\mathbb{E}_{q_t(x_t, x_{t+1}; \phi_{\geq t})} \left[2f_t(x_t; \psi_t)^T[S_t(x_t) - \nabla_{x_t} \log q_t(x_t|x_{t+1}; \phi_t)] - \|f_t(x; \psi_t)\|_2^2\right]$$

$$= \mathbb{E}_{q_t(x_t, x_{t+1}; \phi_{\geq t})} f_t(x_t; \psi_t)^T[2S_t(x_t) - f_t(x_t; \psi_t) - 2\nabla_{x_t} \log q_t(x_t|x_{t+1}; \phi_t)]$$

$$= \mathbb{E}_{q_t(x_t, x_{t+1}; \phi_{\geq t})} [S_t(x_t) - g_t(x_t; \psi_t)]^T[S_t(x_t) + g_t(x_t; \psi_t) - 2\nabla_{x_t} \log q_t(x_t|x_{t+1}; \phi_t)].$$

For maximization w.r.t. $\psi_t$, this target is equivalent to

$$\mathbb{E}_{q_t(x_t, x_{t+1}; \phi_{\geq t})} \left[2f_t(x_t; \psi_t)^T[S_t(x_t) - \nabla_{x_t} \log q_t(x_t|x_{t+1}; \phi_t)] - \|f_t(x; \psi_t)\|_2^2\right]$$

$$= -\mathbb{E}_{q_t(x_t, x_{t+1}; \phi_{\geq t})} \|f_t(x; \psi_t) - S_t(x_t) + \nabla_{x_t} \log q_t(x_t|x_{t+1}; \phi_t)\|_2^2 + C$$

$$= -\mathbb{E}_{q_t(x_t, x_{t+1}; \phi_{\geq t})} \|g_t(x; \psi_t) - \nabla_{x_t} \log q_t(x_t|x_{t+1}; \phi_t)\|_2^2 + C,$$

where $C$ is a term that does not contain $\psi_t$. Therefore, the minimax optimization problem is equivalent to

$$\min_{\phi_t} \quad \mathbb{E}_{q_t(\boldsymbol{x}_t, \boldsymbol{x}_{t+1}; \phi_{\geq t})} \left[\boldsymbol{S}_t(\boldsymbol{x}_t) - \boldsymbol{g}_t(\boldsymbol{x}_t; \psi_t)\right]^T \left[\boldsymbol{S}_t(\boldsymbol{x}_t) + \boldsymbol{g}_t(\boldsymbol{x}_t; \psi_t) - 2\nabla_{\boldsymbol{x}_t} \log q_t(\boldsymbol{x}_t | \boldsymbol{x}_{t+1}; \phi_t)\right],$$

$$\min_{\psi_t} \quad \mathbb{E}_{q_t(\boldsymbol{x}_t, \boldsymbol{x}_{t+1}; \phi_{\geq t})} \|\boldsymbol{g}_t(\boldsymbol{x}_t; \psi_t) - \nabla_{\boldsymbol{x}_t} \log q_t(\boldsymbol{x}_t | \boldsymbol{x}_{t+1}; \phi_t)\|_2^2.$$

### C.3 Joint training of HSIVI

As mentioned in Section 4.2, when parameter sharing scheme is used in the conditional layers for application to diffusion model acceleration, sequential training from $t = T - 1$ to $t = 0$ is not feasible. Therefore, we consider the following training objective

$$\mathcal{L}_{\text{HSIVI-}f}(\phi) = \sum_{t=0}^{T-1} \beta(t) \mathcal{L}_{\text{SIVI-}f}\left(p_t(\boldsymbol{x}_t) \| q_t(\boldsymbol{x}_t; \phi)\right).$$

An intuitive method is to randomly sample a batch of time steps $\{t_k\}_{k=1}^K$ and for each $t_k$ train $\mathcal{L}_{\text{SIVI-}f}\left(p_{t_k}(\boldsymbol{x}_{t_k}) \| q_{t_k}(\boldsymbol{x}_{t_k}; \phi)\right)$ directly. However, sequentially sampling $\boldsymbol{x}_{t_k}$ through $q(\boldsymbol{x}_i | \boldsymbol{x}_{i+1}; \phi)$ from $i = T - 1$ to $i = t_k$ is still necessary in this case, making it memory-consuming to preserve the computation graphs of the entire sampling process.

In order to reduce the cost of accumulating computation graphs, for each $t$, we treat $q_{t+1}(\boldsymbol{x}_{t+1}; \phi)$ as a fixed mixing layer denoted by $\tilde{q}_{t+1}(\boldsymbol{x}_{t+1})$ and only fit the conditional layer $q_t(\boldsymbol{x}_t | \boldsymbol{x}_{t+1}; \phi)$. More specifically, for HSIVI-SM, we consider the following optimization problem

$$\min_{\phi} \sum_{t=0}^{T-1} \beta(t) \mathbb{E}_{\tilde{q}_t(\boldsymbol{x}_t, \boldsymbol{x}_{t+1}; \phi)} \left[\boldsymbol{S}_t(\boldsymbol{x}_t) - \boldsymbol{g}_t(\boldsymbol{x}_t; \psi)\right]^T \left[\boldsymbol{S}_t(\boldsymbol{x}_t) + \boldsymbol{g}_t(\boldsymbol{x}_t; \psi) - 2\nabla_{\boldsymbol{x}_t} \log q_t(\boldsymbol{x}_t | \boldsymbol{x}_{t+1}; \phi)\right], \quad (18)$$

$$\min_{\psi} \sum_{t=0}^{T-1} \beta(t) \mathbb{E}_{\tilde{q}_t(\boldsymbol{x}_t, \boldsymbol{x}_{t+1}; \phi)} \|\boldsymbol{g}_t(\boldsymbol{x}_t; \psi) - \nabla_{\boldsymbol{x}_t} \log q_t(\boldsymbol{x}_t | \boldsymbol{x}_{t+1}; \phi)\|_2^2, \quad (19)$$

where $\tilde{q}_t(\boldsymbol{x}_t, \boldsymbol{x}_{t+1}; \phi) = q_t(\boldsymbol{x}_t | \boldsymbol{x}_{t+1}; \phi) \tilde{q}_{t+1}(\boldsymbol{x}_{t+1})$. In what follows, we demonstrate that the above problem also ensures an accurate approximation of the target score function.

For equation (19), by the denoising score matching trick (Hyvärinen, 2005), the optimal point of $\psi$, denoted by $\psi^*(\phi)$, satisfies

$$\boldsymbol{g}_t(\boldsymbol{x}_t; \psi^*(\phi)) = \nabla_{\boldsymbol{x}_t} \log \tilde{q}_t(\boldsymbol{x}_t; \phi),$$

where $\tilde{q}_t(\boldsymbol{x}_t; \phi) = \int q(\boldsymbol{x}_t | \boldsymbol{x}_{t+1}; \phi) \tilde{q}(\boldsymbol{x}_{t+1}) \mathrm{d}\boldsymbol{x}_{t+1}$. By plugging in the optimal point $\psi^*(\phi)$, each term in equation (18) is equivalent to

$$\mathbb{E}_{\tilde{q}_t(\boldsymbol{x}_t, \boldsymbol{x}_{t+1}; \phi)} \left[\boldsymbol{S}_t(\boldsymbol{x}_t) - \boldsymbol{g}_t(\boldsymbol{x}_t; \psi^*(\phi))\right]^T \left[\boldsymbol{S}_t(\boldsymbol{x}_t) + \boldsymbol{g}_t(\boldsymbol{x}_t; \psi^*(\phi)) - 2\nabla_{\boldsymbol{x}_t} \log q_t(\boldsymbol{x}_t | \boldsymbol{x}_{t+1}; \phi)\right]$$

$$= \mathbb{E}_{\tilde{q}_t(\boldsymbol{x}_t; \phi)} \left[\boldsymbol{S}_t^2(\boldsymbol{x}_t) - \boldsymbol{g}_t^2(\boldsymbol{x}_t; \psi^*(\phi))\right] - 2 \iint \tilde{q}(\boldsymbol{x}_{t+1}) \left[\boldsymbol{S}_t(\boldsymbol{x}_t) - \boldsymbol{g}_t(\boldsymbol{x}_t; \psi^*(\phi))\right]^T \nabla_{\boldsymbol{x}_t} q_t(\boldsymbol{x}_t | \boldsymbol{x}_{t+1}; \phi) \mathrm{d}\boldsymbol{x}_{t+1} \mathrm{d}\boldsymbol{x}_t$$

$$= \mathbb{E}_{\tilde{q}_t(\boldsymbol{x}_t; \phi)} [\boldsymbol{S}_t^2(\boldsymbol{x}_t) - \boldsymbol{g}_t^2(\boldsymbol{x}_t; \psi^*(\phi))] - 2 \int \left[\boldsymbol{S}_t(\boldsymbol{x}_t) - \boldsymbol{g}_t(\boldsymbol{x}_t; \psi^*(\phi))\right]^T \nabla_{\boldsymbol{x}_t} \tilde{q}_t(\boldsymbol{x}_t; \phi) \mathrm{d}\boldsymbol{x}_t$$

$$= \mathbb{E}_{\tilde{q}_t(\boldsymbol{x}_t; \phi)} \left[\boldsymbol{S}_t^2(\boldsymbol{x}_t) - 2\boldsymbol{S}_t(\boldsymbol{x}_t)^T \nabla_{\boldsymbol{x}_t} \log \tilde{q}_t(\boldsymbol{x}_t; \phi) + (\nabla_{\boldsymbol{x}_t} \log \tilde{q}_t(\boldsymbol{x}_t; \phi))^2\right]$$

$$= \mathbb{E}_{\tilde{q}_t(\boldsymbol{x}_t; \phi)} \|\boldsymbol{S}_t(\boldsymbol{x}_t) - \nabla_{\boldsymbol{x}_t} \log \tilde{q}_t(\boldsymbol{x}_t; \phi)\|^2.$$

Therefore, the global optimal point $\phi^*$ also ensures that the score of the variational distribution fits the target score function.

Based on the training objectives (18) (19) mentioned above, we propose Algorithm 2 for joint training, which does not need to store the computation graphs of the sample sequences. Moreover, by assuming an increasing weighting function $\beta(t)$, we assign larger weights $\beta(t)$ for those $t$ close to $T - 1$, which tends to train the conditional layers that are close to $T - 1$ first during the training, resembling the sequential training.

---

**Algorithm 2** Hierarchical semi-implicit variational inference (joint training)

---

**Input:** Auxiliary bridge $\{p_t(\boldsymbol{x})\}_{t=0}^{T-1}$; a weighting function $\beta(t)$; initial value of parameters $\phi^{(0)}$.
**Output:** The optimal parameters $\phi^*$.
Initialization: $\phi \leftarrow \phi^{(0)}$.
**while** not converge **do**
  Uniformly sample $K$ time steps $\{t_k\}_{k=0}^K$ with replacement from $\{0, \ldots, T-1\}$.
  Sample a minibatch $\{\boldsymbol{x}_T^{(k)}\}_{k=1}^K$ from the base distribution $q_T(x)$.
  **for** $k = 1, \ldots, K$ and $t_k < T - 1$ **do**
    Sequentially sample $\boldsymbol{x}_{t_k+1}^{(k)}$ through $q(\boldsymbol{x}_i|\boldsymbol{x}_{i+1}; \phi_i)$ from $i = T - 1$ to $i = t_k + 1$.
    Detach the computation graphs from $\{x_{t_k+1}^{(k)}\}_{k=1}^K$.
  **end for**
  Update $\phi$ by optimizing the objective $\sum_{k=1}^K \beta(t_k)\mathcal{L}_{\text{SIVI-}f}\left(p_{t_k}(\boldsymbol{x}_{t_k})\|q_{t_k}(\boldsymbol{x}_{t_k}; \phi)\right)$, where the $k$-th term is computed based on a single sample $\boldsymbol{x}_{t_k+1}^{(k)}$.
**end while**
$\phi^* \leftarrow \phi$.

---

### C.4 $\epsilon$-training of HSIVI-SM

Another popular formulation of diffusion models is modeling the conditional noise $\boldsymbol{\epsilon}_\theta(\boldsymbol{u}_s, s)$ by optimizing the DDPM loss in equation (14) where $\boldsymbol{u}_s = \sqrt{\alpha(s)}\boldsymbol{u}_0 + \sqrt{1 - \alpha(s)}\boldsymbol{\epsilon}$, introduced as "$\epsilon$-prediction" in Appendix A. Now, let us assume the diffusion bridge is constructed with VP-SDE and we have a pre-trained model of conditional noise $\boldsymbol{\epsilon}^*(\boldsymbol{u}, s)$. Similarly, we construct a sequence of noise models $\{\boldsymbol{\epsilon}_t^*(\boldsymbol{x}_t)\}_{t=0}^{T-1}$ by letting $\boldsymbol{x}_t = \boldsymbol{u}_{s_t}$ and $\boldsymbol{\epsilon}_t^*(\boldsymbol{x}) = \boldsymbol{\epsilon}_t^*(\boldsymbol{x}, s_t)$ which forms a (generalized) $T$-layer diffusion bridge. We only discuss how $\epsilon$-training can be applied to joint training and the derivation for sequential training is similar. In what follows, we consider the transformation of the joint training objective $\mathcal{L}_{\text{HSIVI-SM}}$ for diffusion model acceleration.

By letting the weighting function $\beta(t) = 1 - \alpha(s_t)$ and considering the reparametrization form (17) where $\phi_t$ is replaced by $\phi$, the objective of HSIVI-SM takes the form

$$\mathcal{L}_{\text{HSIVI-SM}}(\phi, \psi) = \sum_{t=0}^{T-1} \mathbb{E}_{\tilde{q}_t(\boldsymbol{x}_t, \boldsymbol{x}_{t+1}; \phi)} \left[ 2\sqrt{\beta(t)}\boldsymbol{f}_t(\boldsymbol{x}_t; \psi)^T[\sqrt{\beta(t)}\boldsymbol{S}_t^*(\boldsymbol{x}_t) + \sqrt{\beta(t)}\boldsymbol{\Sigma}_t^{-1/2}(\boldsymbol{x}_{t+1}; \phi)\boldsymbol{\epsilon})] \right.$$
$$\left. - \|\sqrt{\beta(t)}\boldsymbol{f}_t(\boldsymbol{x}_t; \psi)\|_2^2 \right]. \tag{20}$$

where $\boldsymbol{S}_t^*(\boldsymbol{x}_t)$ is a pre-trained score model. Note that we have $\sqrt{\beta(t)}\boldsymbol{S}_t^*(\boldsymbol{x}_t) = -\boldsymbol{\epsilon}_t^*(\boldsymbol{x}_t)$ by equation (15). Define

$$\tilde{\boldsymbol{f}}_t(\boldsymbol{x}_t; \psi) = \sqrt{\beta(t)}\boldsymbol{f}_t(\boldsymbol{x}_t; \psi),$$
$$\tilde{\boldsymbol{\Sigma}}_t(\boldsymbol{x}_{t+1}; \phi) = \boldsymbol{\Sigma}_t(\boldsymbol{x}_{t+1}; \phi)/\beta(t).$$

The HSIVI-SM objective (20) then takes the form

$$\tilde{\mathcal{L}}_{\text{HSIVI-SM}}(\phi, \psi) = \sum_{t=0}^{T-1} \mathbb{E}_{\tilde{q}_t(\boldsymbol{x}_t, \boldsymbol{x}_{t+1}; \phi)} \left[ 2\tilde{\boldsymbol{f}}_t(\boldsymbol{x}_t; \psi)^T[-\boldsymbol{\epsilon}_t^*(\boldsymbol{x}_t) + \tilde{\boldsymbol{\Sigma}}_t^{-1/2}(\boldsymbol{x}_{t+1}; \phi)\boldsymbol{\epsilon})] - \|\tilde{\boldsymbol{f}}_t(\boldsymbol{x}_t; \psi)\|_2^2 \right] \tag{21}$$

and we call it the objective for $\epsilon$-training. In our implementation of $\epsilon$-training, we directly parametrize $\tilde{\boldsymbol{f}}_t(\boldsymbol{x}_t; \psi)$ and $\tilde{\boldsymbol{\Sigma}}_t(\boldsymbol{x}_{t+1}; \phi)$ instead of $\boldsymbol{f}_t(\boldsymbol{x}_t; \psi)$ and $\boldsymbol{\Sigma}_t(\boldsymbol{x}_{t+1}; \phi)$. The objective (21) is more numerically stable since the magnitude of $\tilde{\boldsymbol{\Sigma}}_t(\boldsymbol{x}_{t+1}; \phi)$ is generally larger than $\boldsymbol{\Sigma}_t(\boldsymbol{x}_{t+1}; \phi)$.

### C.5 Complexity comparison of HSIVI

For methods that we discussed in SIVI variants, which use a single conditional layer (i.e., T=1) and hence would be much cheaper to sample from than HSIVI-SM that uses multiple layers $T > 1$. For methods that we discussed for diffusion models, the computational complexity would be similar if they had the same $T$. That is because we used the same neural network architecture for the conditional layers in HSIVI and the score nets in diffusion models. We have a comparison of the sampling time of different methods in Figure 13.

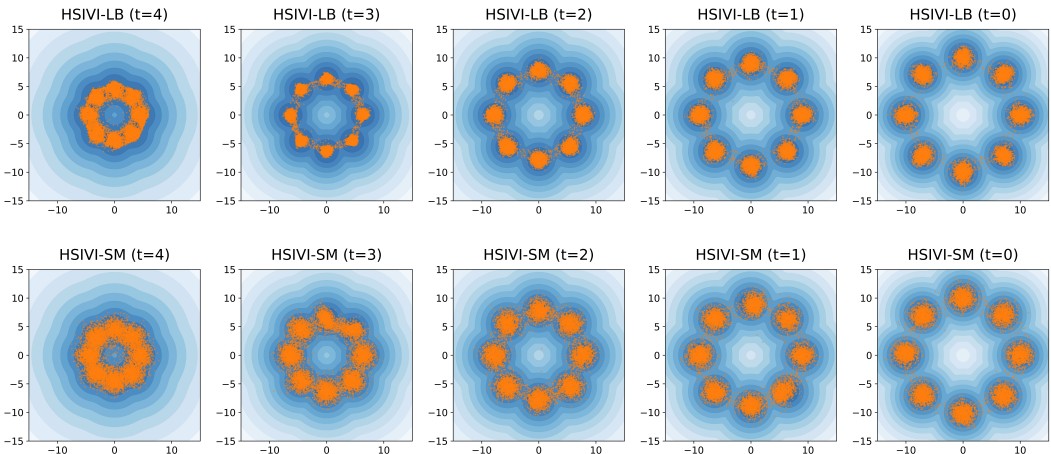

Figure 8: **Upper row**: Sample trajectories progressively generated by 5-layer HSIVI-LB guided by diffusion bridge. **Bottom row**: Sample trajectories progressively generated by 5-layer HSIVI-SM guided by diffusion bridge.

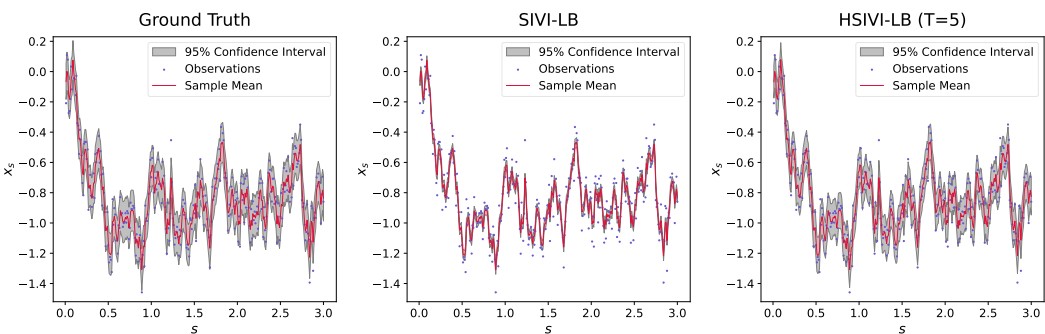

Figure 9: The posterior estimates for conditioned diffusion obtained by SIVI-LB and 5-layer HSIVI-LB. For each method, we collect 100,000 samples to calculate the sample mean and confidence interval.

## D    Additional results of experiments

### D.1    Gaussian mixture model

For HSIVI on the Gaussian mixture model, the auxiliary distributions can also be constructed with diffusion bridge in Example 2. Concretely, the diffusion bridge is constructed by

$$\boldsymbol{x}_t | \boldsymbol{x}_0 \sim \mathcal{N}(\sqrt{\alpha_t}\boldsymbol{x}_0, (1-\alpha_t)\mathbf{I}), \quad \boldsymbol{x}_0 \sim p_0(\boldsymbol{x}_0).$$

where $\alpha_t = \alpha(s_t)$ with $\alpha(s)$ defined in equation (12). In this example, the score function $\boldsymbol{S}_t(\boldsymbol{x}_t) = \nabla_{\boldsymbol{x}_t} \log p_t(\boldsymbol{x}_t)$ has an analytical form

$$\boldsymbol{S}_t(\boldsymbol{x}_t) = \boldsymbol{S}_0 \left( \boldsymbol{x}_t; \sqrt{\alpha_t}\boldsymbol{\mu}, (\alpha_t \sigma^2 + 1 - \alpha_t)\mathbf{I} \right), \quad 0 \leq t \leq T-1.$$

where $\boldsymbol{S}_0(\boldsymbol{x}; \boldsymbol{\mu}, \sigma^2 \mathbf{I})$ is the score function of the Gaussian mixture model $p(\boldsymbol{x}; \boldsymbol{\mu}, \sigma^2 \mathbf{I}) = \sum_{i=1}^{8} 1/8 \cdot \mathcal{N}(\boldsymbol{x}; \boldsymbol{\mu}_i, \sigma^2 \mathbf{I})$. We set the number of layers $T = 5$ and $\alpha_t = 1 - t/5$ for $t = 0, \ldots, 4$. Figure 8 shows the sample trajectories generated by HSIVI. We see clearly that semi-implicit distributions are guided toward the target distribution following the diffusion bridge.

### D.2    High-dimensional conditioned diffusion

We also test SIVI-LB and HSIVI-LB for fitting the posterior in high-dimensional conditioned diffusion. The auxiliary bridge is formed using the same geometric interpolation as for HSIVI-SM,

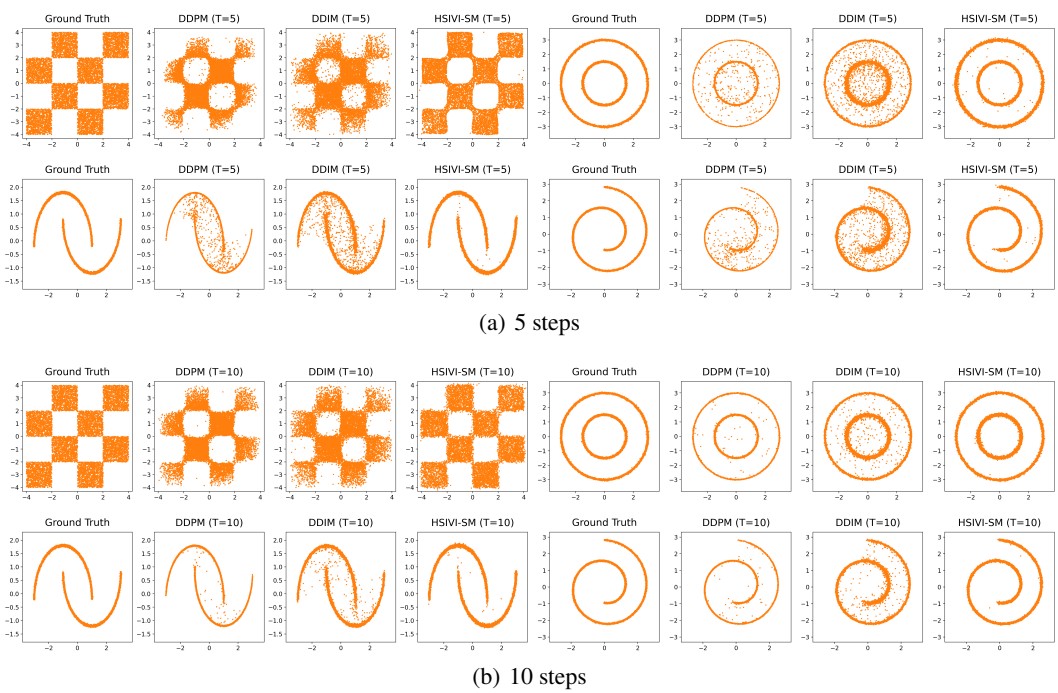

(a) 5 steps

(b) 10 steps

Figure 10: Comparison of 10,000 samples generated by DDPM, DDIM, and HSIVI-SM.

Table 4: Frobenius distances between the estimated covariance matrices and that of the ground truth. For each method, we collect 100,000 samples to estimate the covariance matrix.

|  | $T = 1$ | $T = 2$ | $T = 3$ | $T = 5$ |
| --- | --- | --- | --- | --- |
| HSIVI-SM | 0.0886 | 0.0813 | 0.0431 | **0.0333** |
| HSIVI-LB | 0.0883 | 0.0825 | 0.0722 | **0.0433** |

i.e.

$$p_{\text{base}} = \mathcal{N}(\boldsymbol{x}; \boldsymbol{y}, \sigma^2 \mathbf{I}), \quad \lambda_t = 1 - \frac{t}{T-1} \ \text{ for } \ 0 \leq t \leq T - 1.$$

From Figure 9, we see that SIVI-LB also underestimates the posterior variance and 5-layer HSIVI-LB fits the variance better. This phenomenon is also observed in the performances of SIVI-SM and HSIVI-SM in Figure 3. The quantitative comparison between different numbers of layers is reported in Table 4, where we see that for both HSIVI-SM and HSIVI-LB, the variational approximation gets more accurate with more layers. We also find that HSIVI-SM fits better than HSIVI-LB consistently.

### D.3 Toy examples of diffusion model acceleration

We compare the samples from DDPM, DDIM, and our proposed HSIVI-SM with 5 and 10 steps in Figure 10. We find that DDIM and DDPM fail to converge to the target distribution with a small number of steps, while HSIVI-SM can provide noticeably better samples. Moreover, DDPM tends to underestimate the variance as evidenced by the narrower region occupied by the samples.

### D.4 MNIST

Figure 11 shows the samples from DDPM, DDIM, and HSIVI-SM with $T = 10$ steps. We see that the samples produced by HSIVI-SM is much cleaner and more recognizable than those produced by DDPM and DDIM.

| DDPM (T=10) | DDIM (T=10) | HSIVI-SM (T=10) |
|:---:|:---:|:---:|

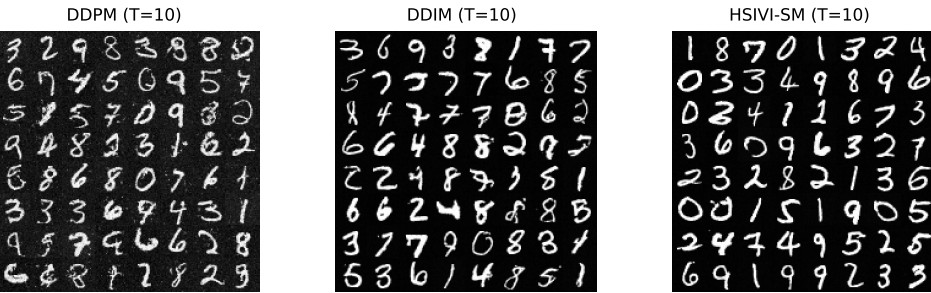

Figure 11: Comparison of the quality of uncurated samples generated by DDPM, DDIM, and HSIVI-SM with 10 discrete time steps on MNIST.

Table 5: Number of parameters in the score model (or noise model) used by different methods in Table 2. 'M' refers to million.

|  | CIFAR-10 | CelebA |
|---|:---:|:---:|
| other methods | 38.72M | 78.66M |
| HSIVI-SM (ours) | 38.72M | 78.66M |

### D.5 CIFAR-10 & CelebA

Figure 12 shows the uncurated samples from our proposed HSIVI-SM method with different numbers of layers on CIFAR-10 ($28 \times 28$), CelebA ($64 \times 64$) and ImageNet ($64 \times 64$). We also compare the sampling time of different methods when NFE $= 5$ in Figure 13.

One can observe that HSIVI-SM has almost the same running time as the simplest DDIM algorithm. Finally, we report the number of parameters in the score model (or noise model) used by different methods in Table 5, which corresponds to Table 2 and Figure 13. In our implementations of HSIVI-SM, the number of parameters in the noise model equals that in the conditional layer $q_t(\boldsymbol{x}_t|\boldsymbol{x}_{t+1}; \phi)$. We find that our model with the same parameters reaches better results in Table 2.

## E   Experimental details

### E.1   Target distribution approximation

In this part, we set the conditional layer to be $q_\phi(\boldsymbol{x}|\boldsymbol{z}) = \mathcal{N}(\boldsymbol{x}; \boldsymbol{\mu}(\boldsymbol{z}; \phi^\mu), \mathrm{diag}\{\exp(\phi^\sigma)\})$ and the mixing layer to be $\mathcal{N}(\boldsymbol{0}, \mathbf{I})$ for SIVI. Here, $\{\phi^\mu, \phi^\sigma\} = \phi$ are the variational parameters. For $T$-layer hierarchical semi-implicit variational distribution with $T \geq 2$, the variational prior $q_T(\boldsymbol{x}_T)$ is set to be $\mathcal{N}(\boldsymbol{0}, \mathbf{I})$. Each conditional layer $q_t(\boldsymbol{x}_t|\boldsymbol{x}_{t+1}; \phi_t)$ for $t = 0, \dots, T-1$ is a conditional Gaussian distribution

$$q_t(\boldsymbol{x}_t|\boldsymbol{x}_{t+1}; \phi_t) = \mathcal{N}(\boldsymbol{x}_t; \boldsymbol{\mu}(\boldsymbol{x}_{t+1}; \phi_t^\mu), \mathrm{diag}\{\exp(\phi_t^\sigma)\}).$$

Note that the $\phi^\sigma$ and $\{\phi_t^\sigma\}_{t=0}^{T-1}$ above are all vectors with the same dimension as $\boldsymbol{x}$. We use sequential training for HSIVI in the two experiments in this part. The parameters $\{\phi_t\}_{t=0}^{T-1}$ are independent across different $t$. If not otherwise specified, we use the Adam optimizer (Kingma & Ba, 2015) with $\beta = (0.9, 0.99)$ for training.

#### E.1.1   Gaussian mixture model

For the experiment on the Gaussian mixture model, we construct 5-layer hierarchical semi-implicit variational distributions. The mean of each conditional layer $\boldsymbol{\mu}(\boldsymbol{z}; \phi^\mu)$ in SIVI or $\boldsymbol{\mu}(\boldsymbol{x}_{t+1}; \phi_t^\mu)$ in HSIVI has a residual form, i.e. $\boldsymbol{\mu}(\boldsymbol{z}; \phi^\mu) = \boldsymbol{z} + \bar{\boldsymbol{\mu}}(\boldsymbol{z}; \phi^\mu)$ and $\boldsymbol{\mu}(\boldsymbol{x}_{t+1}; \phi_t^\mu) = \boldsymbol{x}_{t+1} + \bar{\boldsymbol{\mu}}(\boldsymbol{x}_{t+1}; \phi_t^\mu)$, for $t = 0, \dots, T-1$. $\bar{\boldsymbol{\mu}}(\boldsymbol{z}; \phi^\mu)$ in SIVI and $\{\bar{\boldsymbol{\mu}}(\boldsymbol{x}_{t+1}; \phi_t^\mu)\}_{t=0}^4$ in HSIVI all have the same structures of multi-layer perceptrons (MLPs) with layer widths $[2, 50, 50, 2]$ and ReLU activation functions. For each $t$, $\boldsymbol{f}_t(\boldsymbol{x}_t; \psi_t)$ in HSIVI-SM and $\boldsymbol{f}(\boldsymbol{x}; \psi)$ in SIVI-SM are parameterized by MLPs with layer widths $[2, 128, 128, 2]$ and ReLU activation functions.

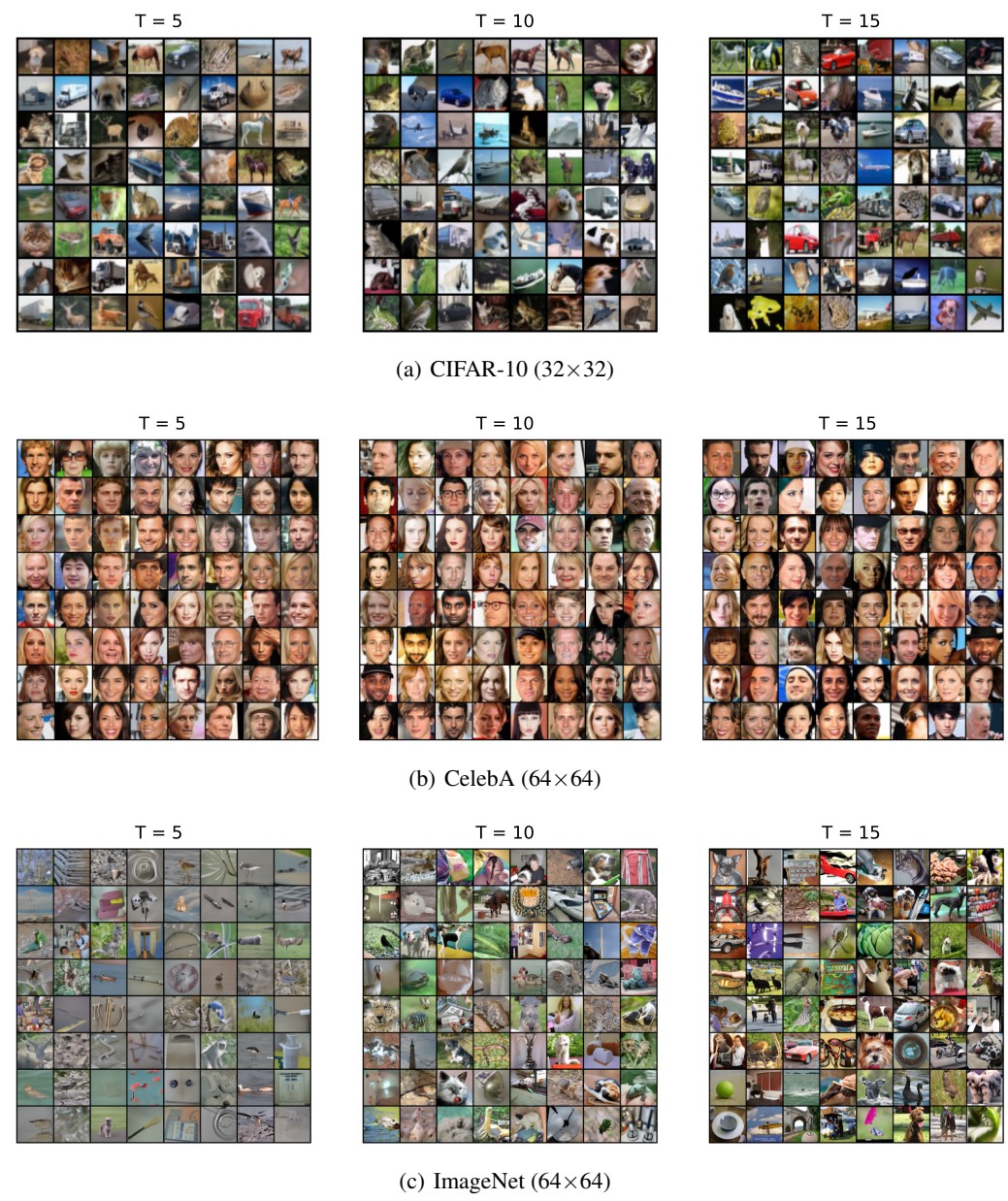

(a) CIFAR-10 (32×32)

(b) CelebA (64×64)

(c) ImageNet (64×64)

Figure 12: Uncurated samples generated by HSIVI-SM with different numbers of layers on CIFAR-10, CelebA and ImageNet.

The noise levels in the diffusion bridge are $1 - \alpha(s_t) = 1 - t/5$ for $t \in \{0, 1, \cdots, 4\}$. We set the learning rate of variational parameters $\phi_t$ (or $\phi$) to 0.001 and the learning rate of $\psi_t$ (or $\psi$) to 0.002 in both SIVI and HSIVI. For HSIVI-LB and HSIVI-SM, we run 80000 variational parameter updates for every conditional layer; for SIVI-LB and SIVI-SM, we run 5×80000 variational parameter updates. For HSIVI-SM and SIVI-SM, in each nested training loop of $\boldsymbol{f}_t(\boldsymbol{x}_t; \psi_t)$ (or $\boldsymbol{f}(\boldsymbol{x}; \psi)$), we update $\psi_t$ (or $\psi$) one time after each update of $\phi_t$ (or $\phi$). All the algorithms are trained with a batch size of 64.

### E.1.2 High-dimensional conditioned diffusion

For the experiment on high-dimensional conditioned diffusion, we examine the performances of SIVI and 5-layer HSIVI. The ground truth is formed by running 100,000 independent stochastic gradient Langevin dynamics (SGLD) chains with a step size of 0.0001 and collecting the results

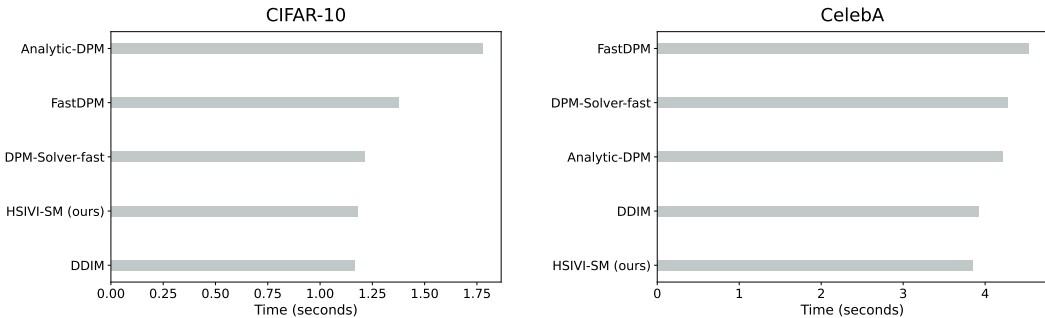

Figure 13: Sampling time ($\downarrow$) of different methods when NFE $= 5$ on CIFAR-10 and CelebA. Results are averaged by 100 independent runs with a batch size of 128 on a single Nvidia 2080Ti GPU.

after 10,000 iterations. For $t = 0, \ldots, T-2$, the mean of each conditional layer $\boldsymbol{\mu}(\boldsymbol{x}_{t+1}; \phi_t^\mu)$ in HSIVI has a residual form, i.e. $\boldsymbol{\mu}(\boldsymbol{x}_{t+1}; \phi_t^\mu) = \boldsymbol{x}_{t+1} + \bar{\boldsymbol{\mu}}(\boldsymbol{x}_{t+1}; \phi_t^\mu)$. For SIVI and $t = T-1$ in HSIVI, we assume $\boldsymbol{\mu}(\boldsymbol{z}; \phi^\mu) = \bar{\boldsymbol{\mu}}(\boldsymbol{z}; \phi^\mu)$ and $\boldsymbol{\mu}(\boldsymbol{x}_{t+1}; \phi_t^\mu) = \bar{\boldsymbol{\mu}}(\boldsymbol{x}_{t+1}; \phi_t^\mu)$. For each $t$, $\bar{\boldsymbol{\mu}}_t(\boldsymbol{x}; \phi_t^\mu)$ in HSIVI and $\bar{\boldsymbol{\mu}}(\boldsymbol{z}; \phi^\mu)$ in SIVI are MLPs with layer widths $[300, 512, 512, 300]$ and ReLU activation functions. For each $t$, $\boldsymbol{f}_t(\boldsymbol{x}_t; \psi_t)$ in HSIVI-SM and $\boldsymbol{f}(\boldsymbol{x}; \psi)$ in SIVI-SM are MLPs with layer widths $[300, 512, 512, 300]$ and ReLU activation functions. For both SIVI and HSIVI, we train each conditional layer for 100,000 iterations with a batch size of 128. For HSIVI-SM and SIVI-SM, in each nested training loop of $\boldsymbol{f}_t(\boldsymbol{x}_t; \psi_t)$ (or $\boldsymbol{f}(\boldsymbol{x}; \psi)$), we update $\psi_t$ (or $\psi$) one time after each update of $\phi_t$ (or $\phi$). We set the learning rate to be 0.0001 for $\phi_t$ (or $\phi$) and 0.0005 for $\psi_t$ (or $\psi$).

### E.2 Diffusion model acceleration

In this part, we use the diffusion bridge to construct the auxiliary distributions and joint training as mentioned in Section 4.2. With a pre-trained score model or noise model, we consider the generative tasks as score-based variational inference problems. Therefore, we do not use any training data to train HSIVI-SM.

For HSIVI-SM, the variational prior $q_T(\boldsymbol{x}_T)$ is set to be $\mathcal{N}(\boldsymbol{0}, \boldsymbol{I})$. To avoid the large memory consumption, we use the joint training method where the parameters of the conditional layers $q_t(\boldsymbol{x}_t|\boldsymbol{x}_{t+1}; \phi)$ and $\boldsymbol{f}_t(\boldsymbol{x}_t; \psi)$ are the same across different $t$. The $t$-th conditional layer is a conditional Gaussian distribution

$$q_t(\boldsymbol{x}_t|\boldsymbol{x}_{t+1}; \phi) = \mathcal{N}\left(\boldsymbol{x}_t; \boldsymbol{\mu}_t(\boldsymbol{x}_{t+1}; \phi^\mu), \mathrm{diag}\left(\sigma_t^2 \exp(\phi^\sigma)\right)\right),$$

where $\{\phi^\mu, \phi^\sigma\} = \phi$ are the variational parameters, $\phi^\sigma$ is a vector with the same dimension as $\boldsymbol{x}$, and $\sigma_t$ is a fixed scalar value. We use the generalized inference process in DDIM (Song et al., 2020a) with the noise level $\eta > 0$ to initialize $\boldsymbol{\mu}_t(\boldsymbol{x}_{t+1}; \phi^\mu)$ and determine the value of $\sigma_t$ for each $t$. If not otherwise specified, we use the Adam optimizer (Kingma & Ba, 2015) with $\beta = (0.9, 0.99)$ for training.

For our implementation, we referenced the training code of diffusion model acceleration for our models in the repository from (Dockhorn et al., 2022).

#### E.2.1 Toy examples of diffusion model acceleration

For pre-training the score model $\boldsymbol{S}^*(\boldsymbol{x}, s)$, we consider quadratic noise levels $1 - \alpha(s) = s^2$ for $s \in [0, 1]$. We then train $\boldsymbol{S}^*(\boldsymbol{x}, s)$ on 1000 fixed noise levels $\{1 - \alpha(i/1000)\}_{i=1}^{1000}$ by optimizing the DDPM loss in equation (13) for 200,000 iterations with a learning rate of 0.0003 and a batch size of 100. For constructing the diffusion bridge, we choose $T$ discrete time steps $\{s_t\}_{t=0}^{T-1}$ so that $1 - \alpha(s_t) = [0.01 + (\sqrt{0.8} - 0.1)t/T]^2$ for $t = 0, 1, \ldots, T-1$.

**Model architecture** The model architecture of $\boldsymbol{S}^*(\boldsymbol{x}, s)$ is

$$\boldsymbol{S}^*(\boldsymbol{x}, s) = \mathrm{MLP}^{\mathrm{dec}}\left(\mathrm{MLP}^{\mathrm{embx}}(\boldsymbol{x}) + \mathrm{MLP}^{\mathrm{embt}}(1 - \alpha(s))\right),$$

where $\mathrm{MLP}^{\mathrm{dec}}$ is a decoder implemented as MLPs with layer widths $[128, 128, 128, 2]$, $\mathrm{MLP}^{\mathrm{embx}}$ is a data embedding block implemented as MLPs with layer widths $[2, 128]$, and $\mathrm{MLP}^{\mathrm{embt}}$ is a time

embedding block implemented as MLPs with layer widths $[256, 128, 128]$. We use the sinusoidal positional embedding (Vaswani et al., 2017) of $1 - \alpha(s)$ as the input of $\text{MLP}^{\text{embt}}$. All these three MLPs use GELU as activation functions. We use the generalized inference process with noise level $\eta = 1.0$ to initialize the conditional layers. The architecture of $\boldsymbol{f}_t(\boldsymbol{x}_t; \psi)$ is the same as that of $\boldsymbol{S}^*(\boldsymbol{x}, s)$. We initialize $\boldsymbol{f}_t(\boldsymbol{x}_t; \psi)$ with $\boldsymbol{S}_t^*(\boldsymbol{x}_t) := \boldsymbol{S}^*(\boldsymbol{x}_t, s_t)$.

**Training setting**    The learning rate is set to be 0.0002 for $q_t(\boldsymbol{x}_t | \boldsymbol{x}_{t+1}; \phi)$ and 0.0005 for $\boldsymbol{f}_t(\boldsymbol{x}_t; \psi)$ on Swissroll, Circles, and Moons for both $T = 5, 10$. On Checkerboard, the learning rate is set to be 0.00001 (0.00002) for $q_t(\cdot | \boldsymbol{x}_{t+1}; \phi)$ and 0.00005 (0.0001) for $\boldsymbol{f}_t(\boldsymbol{x}_t; \psi)$ when $T = 5$ ($T = 10$). We train HSIVI-SM for 25,000 iterations with a batch size of 64 in all cases. In each nested training loop of $\boldsymbol{f}_t(\boldsymbol{x}_t; \psi)$, we update $\psi$ 3 times after each update of $\phi$.

### E.2.2   MNIST

For the experiment on MNIST, we use the pre-trained noise model $\epsilon^*(\boldsymbol{x}, s)$ and train HSIVI-SM with $\epsilon$-training introduced in Section C.4. The following construction of noise schedule comes from Song et al. (2020a). Let $\beta_j = \beta_{\min} + \frac{\beta_{\max} - \beta_{\min}}{999} j$ for $j = 0, \dots, 999$, where $\beta_{\min} = 0.0001, \beta_{\max} = 0.02$. We pre-train the noise model on the 1000 fixed noise levels $1 - \alpha(s) := \prod_{j=0}^{s} \beta_j$ for $s = 0, \dots, 999$ by equation (14). The noise model is trained for 100,000 iterations with a learning rate of 0.0001 and a batch size of 64. We then choose $T$ discrete time steps $s_t = \lfloor 800 \cdot \frac{t^2}{T^2} \rfloor$ for $t = 0, \dots, T - 1$ to construct the $T$-layer diffusion bridge.

**Model architecture**    The pre-trained noise model $\epsilon^*(\boldsymbol{x}, s)$ follows the UNet structure employed by Ho et al. (2020) where the number of input channels and output channels is reduced to one. Additionally, we pad the image size to $32 \times 32$ to fit $\epsilon^*(\boldsymbol{x}, s)$. We use the generalized inference process with noise level $\eta = 0.2$ to initialize the conditional layers. The architecture of $\boldsymbol{f}_t(\boldsymbol{x}_t; \psi)$ is the same as that of $\epsilon^*(\boldsymbol{x}, s)$. We initialize $\boldsymbol{f}_t(\boldsymbol{x}_t; \psi)$ with $-\epsilon^*(\boldsymbol{x}_t, s_t)/\sqrt{1 - \alpha(s_t)}$.

**Training setting**    For both $T = 5, 10$, the learning rate is set to be $1.6 \times 10^{-5}$ for $\phi$ and $6.4 \times 10^{-5}$ for $\psi$. We train HSIVI-SM for 10,000 iterations with a batch size of 64 in all cases. In each nested training loop of $\boldsymbol{f}_t(\boldsymbol{x}_t; \psi)$, we update $\psi$ 20 times after each update of $\phi$.

### E.2.3   CIFAR-10, CelebA & ImageNet

For experiments on CIFAR-10 and CelebA, we use the pre-trained noise model $\epsilon^*(\boldsymbol{x}, s)$ and train HSIVI-SM with $\epsilon$-training introduced in Section C.4. We use the same noise schedule as in the experiment on MNIST. Let $\beta_j = \beta_{\min} + \frac{\beta_{\max} - \beta_{\min}}{999} j$ for $j = 0, \dots, 999$, where $\beta_{\min} = 0.0001, \beta_{\max} = 0.02$. We take the pretrained noise model for CIFAR10 and ImageNet seperately from `https://github.com/tqch/ddpm-torch/releases/download/checkpoints/cifar10_2040.pt` and `https://openaipublic.blob.core.windows.net/diffusion/march-2021/imagenet64_uncond_100M_1500K.pt`. On CelebA, We pre-train the noise model on the 1000 fixed noise levels $1 - \alpha(s) := \prod_{j=0}^{s} \beta_j$ for $s = 0, \dots, 999$ by optimizing equation (14). The noise model is trained for 600 epochs, with a learning rate of 0.00002 and batch size of 128. We then choose $T$ discrete time steps $s_t = \lfloor 800 \cdot \frac{t^2}{T^2} \rfloor$ for $t = 0, \dots, T - 1$ to construct the $T$-layer diffusion bridge.

**Model architecture**    On CIFAR-10 and CelebA, the structure of $\epsilon^*(\boldsymbol{x}, s)$ is exactly the UNet[3] employed in Ho et al. (2020) without modification; on ImageNet, the structure of $\epsilon^*(\boldsymbol{x}, s)$ is exactly the UNet in Nichol & Dhariwal (2021). We use the generalized inference process with noise level $\eta = 0.2$ to initialize the conditional layers. The architecture of $\boldsymbol{f}_t(\boldsymbol{x}_t; \psi)$ is the same as that of $\epsilon^*(\boldsymbol{x}, s)$. We initialize $\boldsymbol{f}_t(\boldsymbol{x}_t; \psi)$ with $-\epsilon^*(\boldsymbol{x}_t, s_t)/\sqrt{1 - \alpha(s_t)}$.

**Training setting**    The number of layers, which is also the number of function evaluations (NFE), is set to be $T = 5, 10, 15$ in our test cases. On CIFAR-10, the learning rate is set to be $1.6 \times 10^{-5}$ for $q_t(\cdot | \boldsymbol{x}_{t+1}; \phi)$ and $8 \times 10^{-5}$ for $\boldsymbol{f}_t(\boldsymbol{x}_t; \psi)$; on CelebA, the learning rate is set to be $1.2 \times 10^{-6}$ for $q_t(\cdot | \boldsymbol{x}_{t+1}; \phi)$ and $6 \times 10^{-6}$ for $\boldsymbol{f}_t(\boldsymbol{x}_t; \psi)$; on ImageNet, the learning rate is set to be $1 \times 10^{-5}$

---

[3] We use the Pytorch implementation of UNet structure in https://github.com/tqch/ddpm-torch.

for $q_t(\cdot|\boldsymbol{x}_{t+1};\phi)$ and $5 \times 10^{-5}$ for $\boldsymbol{f}_t(\boldsymbol{x}_t;\psi)$. We trained HSIVI-SM for 10,000 iterations with a batch size of 128. During each nested training loop of $\boldsymbol{f}_t(\boldsymbol{x}_t;\psi)$, we update $\psi$ 20 times after each update of $\phi$, since we find $\boldsymbol{f}_t(\boldsymbol{x}_t;\psi)$ needs more training empirically to provide reliable guidance. For $T = 10, 15$, we use the above training settings; for $T = 5$, we find that further fine-tuning on the well-trained 15-layer HSIVI-SM for 1,000 iterations yields better results, and we utilize this strategy to optimize the 5-layer HSIVI-SM with a $0.1\times$ smaller learning rate. Experiments need about 1.5 days on CIFAR-10, need about 3 days on CelebA and 4 days on ImageNet using 8 Nvidia 2080 Ti GPUs. During the training, we find that HSIVI-SM converges in the first 30% iterations on CIFAR-10 and converges in the first 50% iterations on CelebA.

# F    Limitations

For the application of accelerating the sampling process of diffusion models, our HSIVI-SM training involves three models: the score model (or noise model), the conditional layers $q_t(\boldsymbol{x}_t|\boldsymbol{x}_{t+1};\phi)$, and $\boldsymbol{f}_t(\boldsymbol{x}_t;\psi)$. As a result, HSIVI-SM requires higher memory consumption due to the involvement of multiple models. Additionally, since our HSIVI algorithm approximates the target distribution using the score function, it necessitates a pre-trained score model (or noise model) with high accuracy and additional training steps. Finally, we recognize that the alternative method HSIVI-LB remains unexplored for accelerating the diffusion model, and we defer this aspect to future research.

