# OpenReview forum: "Hierarchical Semi-Implicit Variational Inference with Application to Diffusion Model Acceleration"
_NeurIPS.cc/2023/Conference — NeurIPS 2023 poster_

### Official Review · Reviewer_AGbR · 2023-07-07

**Soundness:** 4 excellent
**Presentation:** 3 good
**Contribution:** 1 poor
**Rating:** 3
**Confidence:** 2

**Summary:**

The paper introduces hierarchical semi-implicit variational inference (HISIVI) as an extension of semi-implicit variational inference (SIVI). HISIVI incorporates interpolating distributions between prior and target data to train conditional layers progressively, resulting in accelerated sampling in diffusion models.

**Strengths:**

The paper is well-structured and easy to follow.

**Weaknesses:**

- The novelty of HISIVI compared to related work is not clearly demonstrated. It is recommended to include more related works and comparison: cascaded diffusion models[1], diffusion schrodinger bridge[2], flow matching[3], or rectified flow[4], etc.
- The experiment section is relatively weak. Comparisons could strengthen the evaluation (aforementioned references and more [5],[6], etc.)

[1] Ho, J., Saharia, C., Chan, W., Fleet, D.J., Norouzi, M. and Salimans, T., 2022. Cascaded diffusion models for high fidelity image generation. The Journal of Machine Learning Research, 23(1), pp.2249-2281.
[2] De Bortoli, V., Thornton, J., Heng, J. and Doucet, A., 2021. Diffusion Schrödinger bridge with applications to score-based generative modeling. Advances in Neural Information Processing Systems, 34, pp.17695-17709.
[3] Lipman, Y., Chen, R.T., Ben-Hamu, H., Nickel, M. and Le, M., 2022. Flow matching for generative modeling. arXiv preprint arXiv:2210.02747.
[4] Liu, X., Gong, C. and Liu, Q., 2022. Flow straight and fast: Learning to generate and transfer data with rectified flow. arXiv preprint arXiv:2209.03003.
[5] Xiao, Z., Kreis, K. and Vahdat, A., 2021. Tackling the generative learning trilemma with denoising diffusion gans. arXiv preprint arXiv:2112.07804.
[6] Zheng, H., Nie, W., Vahdat, A., Azizzadenesheli, K. and Anandkumar, A., 2022. Fast sampling of diffusion models via operator learning. arXiv preprint arXiv:2211.13449.

**Questions:**

How does the proposed optimization for score function in HISIVI outperform or differ from the conventional implicit score matching or denoising score matching[7]?

[7] Vincent, P., 2011. A connection between score matching and denoising autoencoders. Neural computation, 23(7), pp.1661-1674.

**Limitations:**

This work only deals with the geometric interpolation of distributions without providing justification. Exploring the impact of choice of interpolation on generation would enhance the quality of this work.

---

> ### Author Rebuttal · Authors · 2023-08-09
>
> Thank you for your valuable feedback! Here are our response to your concerns.
>
> Q1:  The novelty of HSIVI compared to related work is not clearly demonstrated. It is recommended to include more related works and comparison: cascaded diffusion models [1], diffusion schrodinger bridge [2], flow matching [3], or rectified flow [4], etc.
>
> A1: First, we'd like to clarify the novelty of HSIVI compared to related works on diffusion model acceleration. (i) HSIVI can be trained without data. As a variational approach, HSIVI uses flexible hierarchical semi-implicit distributions and introduces a sequence of auxiliary distributions for progressive training. For diffusion models, the pre-trained score networks serve as a natural sequence of auxiliary distributions (i.e., diffusion bridge, see Example 2) for HSIVI.
> (ii) Note that most of the current approaches for accelerating diffusion models are ODE-based, which could limit the diversity of generated samples. The HSIVI approach is different from these acceleration techniques by directly accelerating the SDE. We show in Figure 6 that HSIVI can generate diverse samples even from the same starting state.
>
> Thanks for your recommendation! See below for a comparison of HSIVI with these related works.
>
>   - Cascaded diffusion model [1] comprises a cascaded pipeline of diffusion models which generate high-resolution samples by successive upsampling from low-resolution samples. Our HSIVI is clearly distinct from [1] in that we aim at accelerating diffusion sampling by assuming a diffusion bridge between Gaussian prior and data distribution.
>   - Diffusion Schrödinger bridge (DSB) [2] reformulates the generative modeling as a Schrödinger bridge problem in finite time and proposes an approximate IPF algorithm that requires recursively solving the half-bridge problem that matches the joint path distributions, as opposed to HSIVI that fits the marginal distributions.
>   - Flow matching [3] and rectified flow [4] learn ODE based generative models by regressing the vector fields of fixed conditional probability paths (which could be made straight by connecting the samples from the prior and target distributions and constructing a deterministic coupling). HSIVI differs from them in that HSIVI uses pre-trained score networks (and therefore is data-free) and directly accelerates the SDE.
>   - Denoising Diffusion GANs [5] proposes using GANs to model each sampling step and optimizing them through an adversarial loss with data. In constant, we assume an explicit form (e.g., Gaussian) of the conditional layers which is optimized via a VI approach without data.
>   - [6] proposes a neural operator, which learns the trajectories of the probability flow ODE, and allows for high-accuracy simulation in a few steps. This idea can be thought of as a data-driven distillation for ODE. In contrast, HSIVI originates in a variational inference perspective which is trained without data and directly accelerates the SDE.
>
> Q2: The experiment section is relatively weak. Comparisons could strengthen the evaluation (aforementioned references and more [5],[6], etc.)
>
> A2: We compare the HSIVI-SM with more baselines on CIFAR10 (32x32). We want to clarify that as a data-free method, HSIVI-SM inherently faces the difficulty of lacking true samples. Also, due to the extra stochasticity, compressing SDE is generally more challenging than compressing ODE.
>
> Table: Sample quality measured by FID on CIFAR10 (32x32).
> |Model|5| 10| 15|
> |----|----|----|----|
> |DDPM|320.16|278.65|198.0|
> |FastDPM|67.64|9.85|6.1|
> |Analytic-DDPM|93.16|34.54|20.0|
> |Analytic-DDIM|51.86|14.08|8.6|
> |DDIM|41.53|13.73|8.7|
> |DPM-solver-fast|329.13|10.89|4.67|
> |DiffFlow [7] |28.31|22.46 |N/A    |
> |HSIVI-SM|**6.27**|**4.31**|**4.17**|
>
> Table: Sample quality measured by FID on CIFAR10 (32x32) for the other results with different architectures.
> |Model|FID|
> |----|----|
> |FM[3]| 6.35 (142 NFE)|
> |DDGAN[5]|3.75 (4 NFE) |
> |2-Rectified Flow[4] |3.36 (110 NFE)|
> |2-Rectified Flow[4] |12.21 (1 NFE)|
> |2-Rectified Flow(+Distill)[4]|4.85 (1 NFE)|
>
> [7] Diffusion normalizing flow. NeurIPS 2021.
>
> Q3: How does the proposed optimization for score function ... denoising score matching?
>
> A3: As a variational approach, HSIVI, or more specifically, HSIVI-SM that uses a score matching objective (i.e., Fisher divergence), assumes a target score function, instead of estimating it from the data. Therefore, in HSIVI-SM, the score function is fixed, not optimized. However, the semi-implicit variational posterior does not have a tractable score function. To deal with it, HSIVI-SM rewrites the Fisher divergence as the maximum of an inner optimization problem. This allows us to take advantage of the hierarchical structure of semi-implicit distributions and use a similar trick to denoising score matching for a tractable training objective. This technique is borrowed from the work of SIVI-SM (https://openreview.net/forum?id=sd90a2ytrt). We want to emphasize that the improvement of sampling efficiency of HSIVI over denoising diffusion models is not due to the way score matching is done, but the expressiveness of semi-implicit distributions and a training procedure that directly matches the marginal distributions.
>
> Q4: This work only deals with the geometric ...
>
> A4: In addition to the geometric interpolation (Example 1), we also introduced an alternative interpolation method called diffusion bridge (Example 2) which is useful for diffusion models. We also give a comparison of these interpolations in Figure 2 (geometric interpolation results) and Figure 7 in Appendix D.1 (diffusion bridge results) on the Gaussian mixture model. Exploring the impact of the choice of interpolation is interesting. However, for diffusion models, this would also be a bit challenging as it relates to how one constructs the forward process that allows efficient score estimation. We thank the reviewer for your suggestion, and we will leave a more thorough investigation for future work.

---

> > ### Comment · Reviewer_AGbR · 2023-08-15
> >
> > Thank you for addressing all the concerns raised by reviewers.
> > However, due to my limited knowledge, I’m finding it challenging to fully comprehend how HSIVI algorithm works, given the details provided in the paper.
> > Given my current understanding, I find it difficult to distinguish the contributions of this work from those of other generative models.
> > Considering these factors, I have made the decision to lower my confidence score to ensure a fair decision by the ACs.

---

### Official Review · Reviewer_MCSt · 2023-07-07

**Soundness:** 2 fair
**Presentation:** 4 excellent
**Contribution:** 2 fair
**Rating:** 6
**Confidence:** 3

**Summary:**

Semi-implicit variational inference increases the expressiveness of variational posteriors but introduces intractabilities to their inference. This work adds to the semi-implicit variational inference (SIVI) work of Yu and Zhang (https://openreview.net/forum?id=sd90a2ytrt) which uses an alternative objective (the Fisher divergence) that, combined with the hierarchical nature of semi-implicit variational distributions, can be approximated as the solution of a mini-max optimization problem. This work takes inspiration from simulated annealing and denoising-diffusion models by proposing to expand the single-layer semi-implicit variational family into multiple-layer variants by specifying hierarchical auxiliary distributions to guide the semi-implicit variational distribution towards the target distribution. The authors denote this method Hierarchical Semi-Implicit Variational Inference (HSIVI). The auxiliary distributions are specified either by assuming the marginal densities of a geometric interpolation between a target and base distribution are available analytically or by using pre-trained scores from a diffusion bridge. The main contribution to this paper is stated in the author's claim that "when used for diffusion model acceleration, we show that HSIVI can produce high quality samples comparable to or better than the existing fast diffusion model based samplers with small number of function evaluations on various datasets." This claim is tested empirically on the CIFAR-10 (32x32) and CelebA (64x64) datasets with results showing comparable or better FID scores to existing fast diffusion model samplers with few (between 5 and 15) function evaluations. As well as the main contribution, an algorithm for training is provided by noting that the conditional layers can be trained sequentially until convergence by exploiting the hierarchical structure of the semi-implicit distribution. Also, an efficient parameterization of the neural networks is provided to make the algorithm computationally feasible.

**Strengths:**

Quality

The work is a novel combination of well-known, peer-reviewed techniques including simulated annealing and SIVI. It is clear how this work differs from the work of Yu and Zhang (https://openreview.net/forum?id=sd90a2ytrt) using the hierarchical semi-implicit variational posteriors, and the value of this contribution is demonstrated on a multi-modal problem in Figure 2. The methods used seem to work well judging by the experimental results.

Clarity

The submission is well-organized and clearly written.

**Weaknesses:**

Originality

There is limited technical novelty in this paper as the technical content on the work is almost entirely covered by the work of Yu and Zhang (https://openreview.net/forum?id=sd90a2ytrt) that rewrites the Fisher divergence leading to a form that does not require computing the score of the hiearchical variational posterior. The paper extends this by sequentially score matching to noised versions of the target distribution, which have been obtained analytically or approximated from pretrained denoising diffusion models. The main novelty in this work is instead an experimental one, showing that HSIVI can produce high quality samples comparable to or better than the existing fast diffusion model based samplers with a small number of function evaluations on two datasets by comparing FID scores (CIFAR-10 (32x32) and CelebA (64x64)) and one dataset by comparing samples visually (MNIST). The experiments were made computationally feasible by a trick to parameterize the neural networks that is detailed in Proposition 1.

Quality of results

Relatively simple experiments are provided that on one hand demonstrate the contribution well, but on the other hand do not cover a very diverse range of cases. No empirical failure case is provided for a complicated distribution that requires many function evaluations to sample from, which may provide an interesting avenue for future work.

On the reproducibility of the results, whilst the experimental setups are well detailed, no code is provided with the paper, nor indication it will become available upon publication. The reviewer would like to see that the code is made available since this is crucial for reproducibility. Furthermore, there may not be enough information to replicate the experiments. For example, as far as I can see, there is no discussion on the parameterization/values of the positive weighting function, beta(t) for joint HSIVI-SM training on CIFAR-10 and MNIST, which makes it impossible to reproduce without any code provided.


Significance

The work's significance would be improved if it addressed why we should expect improvements in sampling efficiency over denoising-diffusion sampling and where these improvements can be derived from (the particular form of variational family, or something else?).

**Questions:**

1. It would be interesting if the authors could demonstrate a failure case for their hierarchical variational posterior approximating the denoising-diffusion model, what properties of the score in the backwards process (equation 9) would cause the proposed variational posterior to fail with a small number of function evaluations?

2. No interpretation of the neural network f_t(x_t) is given, apart from that it provides "guidance". Could the authors please clarify what is being guided, and to what?

3. Comparing the proposed sampling algorithm to denoising-diffusion: the sampling algorithm for denoising-diffusion uses a reverse-time Markov chain that has specified isotropic variance that appears due to the assumed forward-time transition densities. Whereas the proposed HIVI sampling algorithm in this work is a Markov chain with transition densities that are the conditional variational posterior distributions (called the conditional layer in this work).  The difference between this Markov chain and the denoising-diffusion one is the trainable non-isotropic variance of the conditional layer q_t(\cdot|x_{t+1}; \phi). If the conditional layer is instead modelled as a Gaussian distribution with mean and an isotropic scale-Identity matrix, then do you expect that there is no improvement over denoising-diffusion with the same number of steps? It would be good to see an experiment that showed the trade off between sample quality and number of function evaluations using this simpler variational family, which is on par with the denoising-diffusion sampling.


Typos:

Fisher divergence is not symmetric in its arguments and has been incorrectly written with arguments transposed (the expectation is taken over the distribution of the first argument).
See for example how the Fisher divergence is written in the papers that this work has cited
https://arxiv.org/pdf/1602.03253.pdf
or
https://arxiv.org/pdf/1810.03545.pdf

Line 510, 511, 512 there are typos in the norm regularizing the f network (written as f_{t}(x; \psi_t) and should be f_{t}(x_{t}; \psi_t))

**Limitations:**

The limitation discussion picks up on the important points, but it would be good to know if higher dimensional problems were attempted, or problems with more parameters (such as the "huge VP deep continuous-time model (Song et al., 2020b) that has more channels and layers"), and did the authors run into memory issues that were discussed in Appendix F?

An ablation of the variational family for the conditional layers (as described in question 3) comparing directly to the denoising-diffusion sampling remains unexplored, and would be useful to see.

---

> ### Author Rebuttal · Authors · 2023-08-09
>
> Thank you for your detailed and valuable feedback. We will address your concerns in the following aspects:
> ### Weaknesses
> #### Originality
> Answer: First, HSIVI not only extends SIVI by allowing multiple conditional layers which greatly enhances the flexibility but also introduces a sequence of auxiliary distributions for progressive training that further improves stability. Note that this is only possible in the context of semi-implicit variational inference, where the mixing layer is allowed to be implicit, and this sets HSIVI apart from the previous methods like hierarchical variational models [1]. Moreover, using HSIVI for diffusion models is more than just an experimental one, and there are also significant methodological contributions. Note that most of the current approaches for accelerating diffusion models are ODE-based, which could limit the diversity of generated samples. The HSIVI approach is different from these acceleration techniques by directly accelerating the SDE. We show in Figure 6 that HSIVI can generate diverse samples even if the starting state is the same.
>
> [1] Hierarchical variational models. in ICML 2016.
>
> #### Quality of results
> Answer: Thank you for your suggestions. We provide a failure case in Figure 2 of the rebuttal PDF, demonstrating that the HSIVI-SM algorithm fails when the layer number $T$ is small (the distances of successive auxiliary distributions are large) on a checkerboard distribution. In fact, the score function on the checkerboard target is sharp on the boundaries but vanishes elsewhere. Therefore, fitting this target distribution is somewhat challenging.
>
> Regarding reproducibility, the parameterization of beta(t) and other details can be found in Appendix E.2.2. We also have sent an anonymous link of codes to AC.
>
> #### Significance
> Answer: The improvement in sampling efficiency over denoising-diffusion sampling comes from the expressiveness of semi-implicit variational families. More specifically, we have shown (at least experimentally) that using the same neural network architecture as the score nets, the semi-implicit variational distribution can be used to compress many steps of SDE simulations in DDPM by fitting toward the score functions at the corresponding times (i.e. the score functions in the diffusion bridge in Example 2).
>
> Another point of view is from the training objective. Note that DDPM matches the joint distributions by minimizing $KL(p(x_{0:T−1})|q(x_{0:T−1}))$. When $T$ is small, the variational distribution $q(x_{0:T−1})$ may be insufficient to approximate $p(x_{0:T−1})$ well enough, leading to degraded marginal distribution approximations of $q_0(x_0)$ to $p_0(x_0)$. In contrast, HSIVI matches the marginal distributions $q_t(x_t)$ and $p_t(x_t)$ and hence would ensure a better fit for $p\_0(x\_0)$ especially when $T$ is small.
> ### Questions
> Q1: It would be interesting if the authors could demonstrate a failure case ...
>
> A1: Please refer to the answer of Weakness-Quality above.
>
> Q2: No interpretation of the neural network f_t(x_t) is given ...
>
> A2: The role of $f_t(x_t)$ is to approximate $\nabla \log p_t(x_t) - \nabla \log q_t(x_t)$, thereby identifying regions where the current variational distribution $q_t(x_t)$ fits the target marginal distribution $p_t(x_t)$ insufficiently. We visualized the training dynamics of $f_0(x_0)$ on the checkerboard target in Figure 3 of the global rebuttal PDF.
>
> Q3:  It would be good to see ... using this simpler variational family, which is on par with the denoising-diffusion sampling.
>
> A3: We train HSIVI-SM with isotropic conditional layers on par with DDPM on CIFAR-10 (see the following table).
>
> Table: Sampled quality measured by FID on CIFAR10 (32x32).
> |NFE\Model|DDPM|DDIM|HSIVI-SM (isotropic)|HSIVI-SM (non-isotropic)|
> |----|----|----|----|----|
> |5|320.16|41.53|7.33|**6.27**|
> |10|278.65|13.73|4.78|**4.31**|
> |15|198.00|8.78|4.46|**4.17**|
>
> The above results also support the findings discussed in Significance. The improvement of HSIVI-SM over DDPM stems not only from a more expressive variational distribution but also from the direct matching of the marginal distributions.
>
> Q4: Typos
>
> A4: Thanks for catching the typos. We will correct these typos in our revision.
>
> ### Limitations
> Q5: It would be good to know if higher dimensional ... and did the authors run into memory issues that were discussed in Appendix F?
>
> A5: We have implemented HSIVI-SM using the score function in [1] on ImageNet64, which demonstrates that HSIVI can also handle high dimensional problems with large models.
>
> Table: Sampled quality measured by FID on ImageNet (64x64). All these methods employ the same UNet in [1] with 115.47M parameters.
> |Model\NFE|5|10|15|
> |----|----|----|----|
> |DDPM|402.68|358.80|284.00|
> |Analytic-DDPM|-|60.65|45.98|
> |Analytic-DDIM|-|70.62|41.56|
> |DDIM|147.03|42.31|24.85|
> |DPM-solver-fast|402.43|28.96|20.03|
> |**HSIVI-SM (ours)**|**40.43**|**17.67**|**15.49**|
>
> In fact, we also find that HSIVI can benefit from more accurate score networks (larger models).
> The score networks on CelebA (64x64) that we used in the main text is a relatively small (see the footnote in page 8). We have used a larger UNet in [2] and got better results (see the table below).
>
> Table: Sampled quality measured by FID on CelebA (64x64).
> |Model|5|10|15|
> |----|----|----|----|
> |HSIVI-SM(UNet with 38.72M parameters)|8.29|4.95|4.6|
> |HSIVI-SM(UNet with 78.66M parameters)|**6.22**|**3.09**|**2.23**|
>
> In fact, the memory usage in HSIVI-SM during training is about three times of those baseline methods. We did not run into memory issues as the memory usage is still in an acceptable range. More details are presented in our released code.
>
> [1] Improved denoising diffusion probabilistic models. in ICML 2021.\
> [2] Denoising diffusion probabilistic models. in NeurIPS 2020.
>
> Q6: An ablation of the variational family for the conditional layers ... to see.
>
> A6: We have implemented HSIVI with isotropic conditional layers (in A3).

---

### Official Review · Reviewer_EivT · 2023-07-10

**Soundness:** 3 good
**Presentation:** 3 good
**Contribution:** 2 fair
**Rating:** 6
**Confidence:** 4

**Summary:**

Authors propose a hierarchical semi-implicit variational inference framework by extending the existing SIVI. Authors showed that the proposed HSIVI, given pre-trained score networks, can be used to accelerate the sampling process of diffusion models with the score matching objective. The numerical results show more enhanced performance in faithful modeling of complex distributions and diffusion model acceleration.

**Strengths:**

The paper is fairly well-written, clearly motivated and generally addresses an important problem. The extension of SIVI to a hierarchical model, although not very novel, is intuitive and makes sense. I haven't checked the proofs but the overall approach seems to hold up.

**Weaknesses:**

- Extending SIVI to a hierarchical model, by itself, is not very novel. However, the application to diffusion acceleration is interesting.
- The experiments and presented numerical results could be improved. (check next section for more)

**Questions:**

- It is mentioned in the manuscript that HSIVI requires more memory. How much more memory (with respect to data dimension) does it need compared to baselines?
- ImageNet (64x64) is a common benchmark dataset for diffusion models. I would be nice to have the results for that as well.
- How does the proposed model performs compared to Diffusion Normalizing Flows [1] both in terms of target distribution approximation and sample generation.
- [A suggestion] It would be interesting to extend HSIVI on graphical models (in a setup like [2]).



[1] Diffusion Normalizing Flow, Qinsheng Zhang and Yongxin Chen, NeurIPS 202.
[2] Efficient Inference Amortization in Graphical Models using Structured Continuous Conditional Normalizing Flows, Christian Weilbach, 2019.

**Limitations:**

The limitations to some extent have been discussed but more discussion (specially around memory usage) is needed.

---

> ### Author Rebuttal · Authors · 2023-08-09
>
> Thank you for your helpful feedbacks and suggestions! Here are our responses to them.
>
> Q1: Extending SIVI to a hierarchical model, by itself, is not very novel. However, the application to diffusion acceleration is interesting.
>
> A1: Thank you for acknowledging the novelty of the application to diffusion acceleration! We admit that HSIVI is a natural extension of the SIVI framework. However, we also believe that HSIVI's contributions go beyond SIVI in the following aspects:
>
> 1. HSIVI constructs more expressive mixing layers using multi-layer architectures. Standard SIVI only has one conditional layer. We show that this may not be enough for distributions with complicated structures (see Figure 2 and 3 in Section 5.1). By allowing multiple conditional layers, HSIVI further improves the flexibility of semi-implicit distributions.
>
> 2. HSIVI introduces an auxiliary bridge to alleviate the difficulty of fitting the target distribution. Training HSIVI, therefore, can be done in a sequential manner, where the intermediate semi-implicit distributions were pushed towards the target distribution layer after layer. Note that this is only possible in the context of semi-implicit variational inference, where the mixing layer is allowed to be implicit, and this sets HSIVI apart from the previous methods like hierarchical variational models [4]. These auxiliary distributions would also anchor the intermediate semi-implicit distributions $q\_t(x\_t), t=T-1, \ldots, 1$, making the training process more stable.
>
> Q2: The experiments and presented numerical results could be improved. (check next section for more)
>
> A2: We have new results on ImageNet (64x64) that compare favorably to other baselines (see A4 below). We will add these new results and other relevant baseline methods (e.g., Diffusion Normalizing Flows [1]) to the experiments in our revision.
>
> Q3: It is mentioned in the manuscript that HSIVI requires more memory. How much more memory (with respect to data dimension) does it need compared to baselines?
>
> A3: In addition to the score nets as in standard diffusion models, HSIVI requires the conditional layers $q_t(x_t|x_{t+1};\\phi), t=T-1,\\ldots, 0$ and $f_t(x_t;\\psi), t=T-1,\\ldots, 0$ for training, when parameter sharing is applied. These two additional parts both take the same memory consumption as the score nets. Therefore, the memory consumption of HSIVI during training is about three times of those baseline methods. After training, both the score network and $f_t(x_t;\\psi), t=T-1,\\ldots, 0$ can be removed, and only the conditional layers $q_t(x_t|x_{t+1};\\phi), t=T-1,\\ldots, 0$ are needed for sampling from HSIVI. So during sampling, the memory consumption of HSIVI is about the same as the baseline methods.
>
> Q4: ImageNet (64x64) is a common benchmark dataset for diffusion models. I would be nice to have the results for that as well.
>
> A4: We conducted additional experiments on the ImageNet (64x64) dataset, which demonstrated that HSIVI also achieves significant acceleration to diffusion model sampling at 5, 10, and 15 steps.
>
> Table: Sample quality measured by FID(&#8595;) on ImageNet (64x64). All methods employ the same UNet in [3].
> | Model\NFE | 5  | 10 | 15 |
> | ---- | ---- | ---- | ---- |
> | DDPM  | 402.68 | 358.80 | 284.00 |
> | DDIM  | 147.03 | 42.31 | 24.85 |
> | Analytic-DDPM  | N/A | 60.65 | 45.98 |
> | Analytic-DDIM | N/A |  70.62 | 41.56 |
> | DPM-Solver-fast | 402.43 | 28.96 | 20.03 |
> | **HSIVI-SM (ours)** | **40.43** | **17.67** | **15.49** |
>
>
> Q5: How does the proposed model performs compared to Diffusion Normalizing Flows [1] both in terms of target distribution approximation and sample generation.
>
> A5: Thank you for mentioning this relevant work. Unlike standard diffusion models, Diffusion Normalizing Flows (DiffFlow) allow learnable forward SDE as well and are trained by matching the forward and backward SDEs [1]. Compared to standard diffusion models, DiffFlow would require fewer discretization steps and thus has better sampling efficiency. In contrast, HSIVI is more like standard diffusion models that use a fixed forward process, while achieving better sampling efficiency via variational approaches using hierarchical semi-implicit distributions. Therefore, the training of HSIVI only requires a pre-trained score function and can be data-free.
>
> DiffFlow can be used for density estimation and sample generation. As HSIVI is not designed for density estimation, we compare HSIVI to DiffFlow in terms of sample generation on CIFAR-10 as follows:
>
> Table: Sampled quality measured by FID(&#8595;) on CIFAR10 (32x32).
> | NFE  | DDPM| DDIM| DiffFlow | HSIVI-SM   |
> |--------|------|------|------|------------|
> | 5      | 320.16   | 41.53   | 28.31   | **6.27**  |
> | 10     | 278.65   |13.73   | 22.56   | **4.31**  |
>
> We will add these results in our revision.
>
> [1] Zhang, Qinsheng, and Yongxin Chen. "Diffusion normalizing flow." in NeurIPS 2021.
>
> Q6: [A suggestion] It would be interesting to extend HSIVI on graphical models (in a setup like [2]).
>
> A6: Thank you for the suggestion! Extending HSIVI to graphical models is indeed very interesting. We will read it carefully and think about possible extensions.
>
> Q7: The limitations to some extent have been discussed but more discussion (specially around memory usage) is needed.
>
> A7: We will add more discussion on memory usage in our revision, please see A3 for more details.
>
> [1] Zhang, Qinsheng, and Yongxin Chen. "Diffusion normalizing flow." in NeurIPS 2021.\
> [2] Weilbach, Christian, et al. "Efficient inference amortization in graphical models using structured continuous conditional normalizing flows." in AABI 2019.\
> [3] Nichol, Alexander Quinn, and Prafulla Dhariwal. "Improved denoising diffusion probabilistic models." in ICML 2021.\
> [4] Ranganath, Rajesh, Dustin Tran, and David Blei. "Hierarchical variational models." in ICML 2016.

---

> > ### Comment · Reviewer_EivT · 2023-08-17
> >
> > Thanks authors for the response addressing my concerns/questions. Having read (all) reviews/responses, I'm keeping my score as is. Looking forward to see new results/discussions in the revised version.

---

### Official Review · Reviewer_PvNK · 2023-07-11

**Soundness:** 3 good
**Presentation:** 3 good
**Contribution:** 3 good
**Rating:** 6
**Confidence:** 4

**Summary:**

The authors extend semi-implicit variational inference to have multiple layers of latent variables, vastly increasing the. expressiveness of the model. A nice formulation of the asymptotic lower bound plus a training scheme is introduced as well.

**Strengths:**

This paper is an extension that on the surface seems straight forward. But the authors demonstrate that this is far from the case and introduce a novel formulation of the semi-vi ELBO.

The paper was also written well and easy to follow. The experiments section was great too.

**Weaknesses:**

My only qualm is the flow between the first half of the paper and the second half where the focus is on diffusion models. Previously, the authors argued that training using a sequence of distributions would lead to better marginal distributions. But for speeding up sampling of diffusion models, it seems like training the joint distribution leads to impressive results. This opens up the question of whether the sequence of distributions is even needed at akk.

**Questions:**

I don't have any questions.

**Limitations:**

It seems like the main text is missing a limitations section.

---

> ### Author Rebuttal · Authors · 2023-08-09
>
> Thank you for your careful review and valuable feedback! Below are our answers to your comments:
>
> Q1: My only qualm is ... But for speeding up sampling of diffusion models, it seems like training the joint distribution leads to impressive results. This opens up the question of whether the sequence of distributions is even needed at all.
>
> A1: Thank you for the discussion on this issue! We want to clarify a potential confusion here regarding "joint training" and "training the joint distribution" in the diffusion model acceleration part. In our proposed HSIVI approach, only the marginal distributions are fitted, not the joint distributions (we have a related discussion in Section 3.2). The phrase "joint training" is introduced when we use parameter sharing to reduce memory usage which is a common practice in diffusion models (e.g., score nets). Since the conditional layers now are parameterized with the same $\\phi$, they would be trained jointly, instead of being independently trained as in the sequential training case where each layer has its own parameters. Therefore, "joint training" is used to indicate a training behavior and it is the marginal distribution $p_t(x_t), t=0,\\ldots, T$ given by a sequence of auxiliary distributions that are approximated progressively with the hierarchical semi-implicit distributions. We see this marginal approximation property as one of the keys that allow HSIVI to generate high-quality samples with small function evaluations (see the discussion paragraph in section 3.2, i.e. line 150-157, for more details).
>
>
> Q2: It seems like the main text is missing a limitations section.
>
> A2: Due to the space constraints, we have placed the limitations section in the appendix. We will move the limitations section to the main text in our revision if there is room.

---

### Official Review · Reviewer_i7TH · 2023-07-15

**Soundness:** 3 good
**Presentation:** 3 good
**Contribution:** 3 good
**Rating:** 7
**Confidence:** 4

**Summary:**

The paper introduces a hierarchical semi-implicit variational inference that stacks multiple semi-implicit layers to construct a flexible generative distribution. The paper introduces a sequence of distributions that interpolate between the target and a base distribution. Each pair of intermediate inference and target distributions are matched by optimizing a SIVI bound. The method is applied in the diffusion model for sampling acceleration.

##Post-rebuttal

Thanks for the authors' response. Most of my questions have been addressed and I will maintain my current score.

**Strengths:**

- The idea of using semi-implicit distribution for the progressive approximation is interesting and intuitive.

- The empirical results show HSIVI improves the distribution approximation and sample quality significantly on several tasks.

- Though the modeling and training of HSIVI consists of many components, the paper explains the ideas concisely and clearly.

**Weaknesses:**

- The validity of the training procedure needs more concrete explanations or theoretical results

- The empirical study needs more evidence for the improvement of acceleration.

- Some closely related works need more discussions of similarity and improvement.

Please see Questions for more details.

**Questions:**

- To train HSIVI sequentially, the paper mentions, “Let the parameters ϕ_t in the t-th conditional layer be independent across different ts.” But since q_t(x_t; ϕ_t) depends on x_{t-1} that depends on ϕ_{t-1}, It is understandable that the graph detaching in the algorithm benefits computation efficiency. Does it have any downside for not training ϕs jointly? For example, will there be convergence issues, or will some ϕs overfit while others underfit?

- In Appendix C.3., the author shows that the detaching operation keeps the optimality of \phi unchanged in the joint training. Does a similar conclusion apply to the sequential training in the main paper?

- Table 1 shows with the same layers/steps, HSIVI achieves better approximation than DDPM and DDIM. Would the computation with one layer of HSIVI be more than one step of DDPM?  If so, it might not be a fair comparison. Can the author discuss the computational complexity and provide wall clock time?

- [1] and [2] design a hierarchical model using semi-implicit distributions, which seem to be closely related to HSIVI. The paper might need more detailed discussions of these papers.

[1] Importance weighted hierarchical variational inference (NeurIPS 2019)
[2] Structured Semi-Implicit Variational Inference (AABI 2019)

---

> ### Author Rebuttal · Authors · 2023-08-09
>
> Thank you for your careful review and helpful feedback! We address your concerns as follows.
>
> Q1: The validity of the training procedure needs more concrete explanations or theoretical results.
>
> A1: The training procedure is similar to SIVI-SM (https://openreview.net/forum?id=sd90a2ytrt), with the only difference being using multiple auxiliary distributions instead of the target distribution directly.
>
> Q2: The empirical study needs more evidence for the improvement of acceleration.
>
> A2: Thank you for your suggestion! We have provided additional results on CelebA (64x64), ImageNet (64x64) with more powerful pre-trained score nets. These results show that our methods work well for common benchmark datasets for diffusion models and can benefit from a more accurate score function. For CelebA (64x64) experiments here, we adopt **the identical Unet in [3] with 78.66M parameters, which is bigger than the one we used in the main text** with one more downsampling block and one more upsampling block. For ImageNet (64x64), we follow DPM-solver, taking the same improved UNet in [4], which is larger than the one in [3] and has 115.47M parameters.
>
> Table: Sample quality measured by FID on CelebA (64x64). All methods without \* employ the same UNet in [3].
> |Model\NFE|5|10|15|
> |----|----|----|----|
> |DDPM|366.10|309.95|206.92|
> |DDIM|27.38|10.89|7.78|
> |FastDPM|27.63|15.44|12.05|
> |Analytic-DDPM|50.92|28.93|21.84|
> |Analytic-DDIM|29.40|15.74|12.25|
> |DPM-Solver-fast|355.96|6.76|2.98|
> |**HSIVI-SM (ours), in the main text (smaller UNet)\***|8.29|4.95|4.66|
> |**HSIVI-SM (ours)**|**6.22**|**3.09**|**2.23**|
>
> Table: Sample quality measured by FID on ImageNet (64x64). All methods employ the same UNet in [4].
> |Model\NFE|5|10|15|
> |----|----|----|----|
> |DDPM|402.68|358.80|284.00|
> |DDIM|147.03|42.31|24.85|
> |Analytic-DDPM|N/A|60.65|45.98|
> |Analytic-DDIM|N/A|70.62|41.56|
> |DPM-Solver-fast|402.43|28.96|20.03|
> |**HSIVI-SM (ours)**|**40.43**|**17.67**|**15.49**|
>
> [3] Denoising diffusion probabilistic models. NeurIPS 2020.\
> [4] Improved denoising diffusion probabilistic models. ICML 2021.
>
> Q3: Some closely related works need more discussions of similarity and improvement.
>
> A3: We will add discussions on related works in our revision. Please see more details in A7.
>
> Q4: To train HSIVI sequentially ... Does it have any downside for not training $\phi$s jointly? For example, will there be convergence issues, or will some $\phi$s overfit while others underfit?
>
> A4: Thanks for this interesting question! In the current setting, each conditional layer is trained to push the fitted distribution from the previous layers toward the auxiliary distribution at the next time step, and then it is fixed afterward. The reason is that the trained $q_t(x_t; \phi\_{\ge t})$ would provide a good approximation to the corresponding auxiliary distribution $p_t(x_t)$ and hence reduce the difficulty for training the next conditional layers (since the differences between successive auxiliary distributions are assumed to be small). Note that this approach does not require each $q_t(x_t; \phi_{\ge t})$ to be a perfect match of $p_t(x_t)$, as the conditional layer at time $t-1$ would automatically compensate for the error from previous time steps when fitting $p_{t-1}(x_{t-1})$ (see an example in Figure 7 in the appendix). Therefore, we do not expect convergence issues given enough capacity of the conditional layers and an appropriate $T$. On the other hand, when $\phi$s are trained jointly, there is a potential drawback that the previously fitted $q_t(x_t; \phi\_{\ge t})$ would deviate from the corresponding auxiliary distribution, and this would make the training less stable as well (see Figure 1 in the rebuttal PDF). We will add a discussion to our revision.
>
> Q5: In Appendix C.3., the author shows that the detaching operation keeps the optimality ... Does a similar conclusion apply to the sequential training in the main paper?
>
> A5: Yes, this result holds true as well. In sequential training, we assume that the marginal distribution of the variational distribution is fitted to the target distribution $p_{t+1}$. This is used as a mixing layer during the training of $q_t(x_t|x_{t+1};\phi_t)$, where we apply a detaching operation to keep the parameters $\phi^\star_{\ge t+1}$ fixed. As a result, the optimality of the model is maintained until the final $t=0$. We will include these details in Appendix C.3.
>
>
> Q6: Would the computation with one layer of HSIVI be more than one step of DDPM? If so, it might not be a fair comparison. Can the author discuss the computational complexity and provide wall clock time?
>
> A6: The computation with one layer of HSIVI is about the same as one step of DDPM/DDIM as we used the same neural network architecture for the conditional layers in HSIVI and the score nets in diffusion models. A wall clock time comparison is provided in Figure 12 in the appendix.
>
> Q7: [1] and [2] design a hierarchical model using semi-implicit distributions, which seem to be closely related to HSIVI. The paper might need more detailed discussions of these papers.
>
> A7: Thank you for sharing these relevant papers. We will add more detailed discussions of these papers in our revised manuscript. (i) IWHVI [1] employs a reverse model as an importance distribution for the variational distribution and requires an explicit variational prior. Our proposed HSIVI inherits the advantage of SIVI that allows $q_t(x_t;\phi_{\geq t})$ to be implicit and does not require a reverse model. These properties of HSIVI improve its expressiveness and enable multi-layer extension. (ii) Structured SIVI [2] assumes an autoregressive form of variational distributions to factorize high-dimensional joint semi-implicit distribution into the product of low-dimensional conditional semi-implicit distributions. Our approach is distinct from structured SIVI because HSIVI does not factorize the data space but augments it with auxiliary bridges to accommodate the multimodal targets.

---

### Official Review · Reviewer_T15h · 2023-07-18

**Soundness:** 3 good
**Presentation:** 3 good
**Contribution:** 3 good
**Rating:** 6
**Confidence:** 2

**Summary:**

This paper presents a new method called Hierarchical Semi-Implicit Variational Inference (HSIVI) that enhances the expressiveness of Semi-Implicit Variational Inference (SIVI) on complex target distributions. HSIVI works by applying SIVI to a hierarchy of latent variables, and using the variational distribution of the previous step as the implicit prior for the later step. The authors apply HSIVI for sampling from diffusion models, learned on a variety of synthetic and real-world datasets.

**Strengths:**

1. Novelty: The paper presents a new method called Hierarchical Semi-Implicit Variational Inference (HSIVI) that enhances the expressiveness of Semi-Implicit Variational Inference (SIVI) on complicated target distributions. This is a novel approach that has not been explored before, to the best of my knowledge.

2. Effectiveness: The authors demonstrate the effectiveness of HSIVI on a variety of synthetic and real-world datasets, including accelerating the sampling process of diffusion models. The results show that HSIVI outperforms existing methods in terms of accuracy and efficiency.

3. Clarity: The paper provides a detailed explanation of the HSIVI method, including its mathematical formulation and implementation. The authors also provide clear and concise descriptions of the experiments and results.


**Weaknesses:**

The main weakness of this work is that the method seem fairly complicated to implement in a practical setting. The method is presented in a general way, with an algorithm box. However, later another objective function is presented with some parameter sharing. Then, in the larger-scale experiments, HSIVI-SM is trained by fune-tuning a larger model. Either the method is not extremely stable / usable, or there are too many details but those are necessary for the experiment to be setup properly? (in the latter case, perhaps one experiment should go to the supplements?).

**Questions:**

(1) Why isn't the performance of HSIVI-LB reported on the other experiments?
(2) In terms of computational efficiency, it wasn't clear to me that if two compared methods had the same "T", then their computational burden were comparable. Are the computational complexities exactly the same for sampling from HSIVI-SM and SIVI, for equal "T"?
(3) In principle, if HSIVI was used to train the model, we could reach a better data likelihood? Why aren't the authors restricting the application of HSIVI to sampling, and not learning the score function as well?

**Limitations:**

Yes

---

> ### Author Rebuttal · Authors · 2023-08-08
>
> Thank you for your thoughtful review and valuable feedback. We address your specific questions and comments below.
>
> Q1: The main weakness of this work is that the method seem fairly complicated to implement in a practical setting. The method is presented in a general way, with an algorithm box. However, later another objective function is presented with some parameter sharing. Then, in the larger-scale experiments, HSIVI-SM is trained by fine-tuning a larger model. Either the method is not extremely stable / usable, or there are too many details but those are necessary for the experiment to be setup properly? (in the latter case, perhaps one experiment should go to the supplements?).
>
> A1: Thanks for your question!
>
> First of all, we propose HSIVI as a general VI method. When used for diffusion model acceleration, we employed parameter sharing to reduce memory consumption so that HSIVI-SM will have the same memory usage as the other baseline methods (e.g., DDPM and DDIM). Moreover, parameter sharing also allows joint training of the conditional layers which would facilitate convergence, similar to parameter sharing of score nets in diffusion models.
>
> Note that the target data distributions in generative models are usually complicated. When $T$ is small, the distance between the auxiliary distributions $\\{p_i(x)\\}_{i=0}^{T-1}$ would be large, making it challenging to train the conditional layers starting from random initialization. As DDIM can produce reasonably good samples when $T$ is small, it serves as a natural initialization strategy for HSIVI. For the same reason, we used the trained $T=15$ layers HSIVI model to initialize $T=5$ layer HSIVI model. These strategies work well in practice, and we see significant improvement of HSIVI over DDIM (Table 2 in the main text). More details of the experimental setup can be found in Appendix E.2.3. Note that these strategies are more like initialization strategies that can take advantage of current fast diffusion model based samplers and is data free, instead of fune-tuning strategies that often require data samples during training.
>
> We apologize for the confusion and would clarify these in our revision.
>
> Q2: Why isn't the performance of HSIVI-LB reported on the other experiments?
>
> A2: The reason is that only the score function is available in diffusion models, not the probability density function. This makes HSIVI-SM a natural choice as it uses the score matching objective instead of the ELBO related objectives.
>
>
> Q3: In terms of computational efficiency, it wasn't clear to me that if two compared methods had the same "T", then their computational burden were comparable. Are the computational complexities exactly the same for sampling from HSIVI-SM and SIVI, for equal "T"?
>
> A3: First, we'd like to clarify that SIVI uses a single conditional layer (i.e., T=1) and hence would be much cheaper to sample from than HSIVI-SM which uses multiple layers $T>1$. For methods that we discussed for diffusion models, the computational complexity would be similar if they had the same $T$. That is because we used the same neural network architecture for the conditional layers in HSIVI and the score nets in diffusion models. We have a comparison of the sampling time of different methods in Figure 12 in Appendix D.5. We will clarify this in our revision.
>
>
> Q4: In principle, if HSIVI was used to train the model, we could reach a better data likelihood? Why aren't the authors restricting the application of HSIVI to sampling, and not learning the score function as well?
>
> A4: Sorry for the confusion! Generally speaking, HSIVI is a variational inference method that assumes the target density is accessible (e.g., the density function up to a constant or the score function is available). When used for diffusion model acceleration, HSIVI-SM does not directly target the generative model. Instead, it requires a sequence of auxiliary distributions that bridges between a simple distribution and the target distribution which is available given the learned score functions of diffusion models (Example 2). Note that this also allows data-free training of HSIVI-SM to accelerate diffusion model sampling. Learning the score functions together with HSIVI is an interesting idea, and would allow for a better forward process. However, it would also be more challenging as the score functions required by HSIVI now need to be learned together with the conditional layers of hierarchical semi-implicit distributions.

---

> > ### Comment · Reviewer_T15h · 2023-08-16
> > **Thank you for the answers**
> >
> > I thank the authors for answering my questions!

---

### Author Rebuttal · Authors · 2023-08-09

We thank all reviewers for their constructive feedback. First, we'd like to address some common issues raised by the reviewers.

### Novelty/Originality

As a semi-implicit variational inference method, HSIVI further enhances the expressiveness of semi-implicit variational posteriors by allowing multiple conditional layers. This makes HSIVI able to provide an accurate approximation of complicated distributions that would not be approximated well via SIVI which uses a single conditional layer. Moreover, we also introduce a sequence of auxiliary distributions for progressive training that further improves the stability of the algorithm.

For diffusion model acceleration, HSIVI-SM can produce high-quality samples that compare favorably to other baseline methods with small numbers of function evaluations. We want to emphasize that it is the expressiveness of semi-implicit distributions and the variational inference approach that progressively matches a sequence of auxiliary distributions (i.e., the diffusion bridge given by the learned score nets) that contributes to the improved sampling efficiency of HSIVI-SM. Although DDPM is originally derived via a variational approach, it matches the joint distributions instead of the marginal distributions. In HSIVI-SM, the marginal distribution of the backward models (i.e., semi-implicit distributions $q_t(x_t), t=T-1\ldots,0$) and the forward models $p_t(x_t), t=T-1,\\ldots,0$ are directly matched via score matching, and we see this as one of the keys that allow HSIVI-SM to generate high-quality samples with a small function evaluations (see the discussion paragraph in Section 3.2, i.e. line 150-157, for more details). Note that this also allows a direct compression of stochastic diffusion models, which is different from most of the current ODE-based acceleration methods. HSIVI-SM can also be trained without data, which would be useful when there are privacy concerns.

Overall, we think there are significant methodological innovations in HSIVI that make it a novel/original contribution to the community.

### More experiments

We have provided additional results on CelebA (64x64) and ImageNet (64x64) with more powerful pre-trained score nets (bigger models with more parameters). These results show that our methods work well for common benchmark datasets for diffusion models and can benefit from more accurate score function estimation.

Table: Sample quality measured by FID on CelebA (64x64). All methods without \* employ the same UNet in [1].
| Model\NFE | 5  | 10 | 15 |
| ---- | ---- | ---- | ---- |
| DDPM  | 366.10 | 309.95 | 206.92 |
| DDIM  | 27.38 | 10.89 | 7.78 |
| FastDPM  | 27.63 | 15.44 | 12.05 |
| Analytic-DDPM  | 50.92 | 28.93 | 21.84 |
| Analytic-DDIM | 29.40 |  15.74 | 12.25 |
| DPM-Solver-fast | 355.96 |  6.76 | 2.98 |
| **HSIVI-SM (ours), in the main text (smaller UNet)\*** | 8.29 |  4.95 | 4.66 |
| **HSIVI-SM (ours)** | **6.22** | **3.09** | **2.23** |

Table: Sample quality measured by FID on ImageNet (64x64). All methods employ the same UNet in [2].
| Model\NFE | 5  | 10 | 15 |
| ---- | ---- | ---- | ---- |
| DDPM  | 402.68 | 358.80 | 284.00 |
| DDIM  | 147.03 | 42.31 | 24.85 |
| Analytic-DDPM  | N/A | 60.65 | 45.98 |
| Analytic-DDIM | N/A |  70.62 | 41.56 |
| DPM-Solver-fast | 402.43 | 28.96 | 20.03 |
| **HSIVI-SM (ours)** | **40.43** | **17.67** | **15.49** |

We will revise our manuscript in the following aspects:

- We will add these additional results on CelebA (64x64), ImageNet (64x64) to the experiments.


- We will enhance the discussion of HSIVI's computational complexity and memory usage in comparison to the baseline methods.
- We will add the other relevant baseline methods (e.g., Diffusion Normalizing Flows, Rectified Flow) to the experiments and the discussion.
- We will add the ablation studies to discuss:
  - The failure case of HSIVI when the number of layers is small.
  - The effect of the accuracy of score net.
- We will add more discussion on the detaching operation and its effect on the optimality of $\phi$ in Appendix C.3.

[1] Ho, Jonathan, Ajay Jain, and Pieter Abbeel. "Denoising diffusion probabilistic models." in NeurIPS 2020.\
[2] Nichol, Alexander Quinn, and Prafulla Dhariwal. "Improved denoising diffusion probabilistic models." in ICML 2021.

We hope our revision has adequately addressed the reviewers' questions and concerns, and look forward to reading any further comments.

---

### Decision · Program_Chairs · 2023-09-21

**Decision:**

Accept (poster)

**Comment:**

Six reviewers evaluated this submission and the scores are 676663. All reviewers agree that this paper proposes a novel or interesting approach. Some concerns on experimental setup and complexity of practical implementation have been raised and adequately addressed in the rebuttal. The only reviewer who gave negative scores had lowered their confidence score after discussion. Therefore, I recommend acceptance.